# Exposure to high-altitude hypobaric hypoxic environment induces low-frequency hearing loss in C57BL/6J mice: Mediated by slowing down the postsynaptic electrical signal transmission speed in the cochlear-inferior colliculus auditory signaling pathway

**Benhong Ren[1,2,3], Qingping Zhang[1,2], Wenyuan Gan[1,2], Guanghao Yue[1], Shanhong Li[1], Xiaoli Zhang[1], Wenjun Cao[1], Feng Tang[4], Ying Zhang[1], Bin Guo[1]\*, Yi Wang[1,5]\***

**1** Otolaryngology Department, Qinghai University Affiliated Hospital, Xining, Qinghai, China,
**2** Otolaryngology Major, Clinical Medical College, Qinghai University, Xining, Qinghai, China,
**3** Department of Otolaryngology-Head and Neck Surgery, Shanxi Fenyang Hospital, Lvliang, Shanxi, China, **4** Plateau Medical Research Center, Qinghai University, Xining, Qinghai, China, **5** Department of Otolaryngology, Peking Union Medical College Hospital, Chinese Academy of Medical Sciences & Peking Union Medical College, Beijing, China

* wegreatgroup@163.com (YW); guobin.3a@163.com (BG)

## Abstract

### Objective

To reveal the pattern of effects of exposure to high-altitude hypobaric hypoxia environment on the auditory function of C57BL/6J mice, so as to provide a theoretical basis for the targeted prevention and treatment of hearing loss, tinnitus, and otogenic vertigo in high-altitude areas.

### Methods

Forty male C57BL/6J mice were randomly divided into a control group and an experimental group. The experimental group was further subdivided into 9 subgroups according to the exposure duration (3, 5, 7, 10, 15, 20, 25, 30, and 35 days), with 4 mice in each subgroup. Mice in the experimental group were housed in a hypobaric hypoxic chamber simulating an altitude of 6000 meters. Auditory function-related indicators of mice in both groups were detected using auditory brainstem response (ABR), electrocochleography (ECochG), and distortion product otoacoustic emissions (DPOAE). Statistical methods were used to compare the intergroup differences.

### Results

Results of ABR testing: Under click stimulation, the latencies of Waves I, II, III, IV, and V, as well as the I-III interwave interval, were significantly prolonged in all subgroups

**Data availability statement:** All relevant data are within the paper and its Supporting Information files.

**Funding:** Financial Disclosure: This study was supported by the Qinghai Provincial Department of Science and Technology (Grant No.: 2023-SF-129, received by Yi. Wang; official website link: https://kjt.qinghai.gov.cn/content/zw/id/14). The funder had no role in the study design, data collection and analysis, decision to publish, or preparation of the manuscript. No other funding was received for this study.

**Competing interests:** The authors have declared that no competing interests exist.

of the experimental group. Abnormal changes (elevation or reduction) in the amplitudes of Waves II and III were observed, and hearing thresholds were significantly elevated ($P < 0.05$). Under stimulation with low-frequency 4000 Hz and 8000 Hz pure tones, all subgroups of the experimental group also exhibited similar changes as mentioned above ($P < 0.05$). Horizontal comparison of different stimulation conditions (click, 4000 Hz, 8000 Hz) within the experimental group revealed that the degree of hearing loss was more significant in the click group and 4000 Hz pure tone stimulation group than in the 8000 Hz group ($P < 0.05$). Results of ECochG testing: The latencies of the summating potential (-SP) and compound action potential (AP) were generally prolonged in all subgroups of the experimental group. Among them, the -SP/AP amplitude ratio was $> 0.4$ in the subgroups exposed for 3, 5, 15, 20, 25, and 30 days ($P < 0.05$), and the -SP/AP area ratio was $> 2.0$ in the 7-day exposure subgroup. Results of DPOAE testing: Among all 40 mice, only 3 passed the test, with a pass rate of 7.5%, and all of these 3 mice were from the control group.

## Conclusion

Exposure to a high-altitude hypobaric hypoxia environment can induce low-frequency hearing loss in C57BL/6J mice, and the process of auditory response regulation and adaptation of mice to this environmental stimulus exhibits time dependence. Further analysis revealed that this low-frequency hearing loss is closely associated with the slowing of postsynaptic electrical signal transmission speed in the cochlea-inferior colliculus auditory conduction pathway; the mechanism underlying this slowing of electrical signal transmission may be related to changes in voltage or resistance at various sites of the auditory pathway during sound signal conduction. Based on auditory physiological mechanisms, it is hypothesized that the aforementioned abnormal changes in voltage or resistance may be associated with endolymphatic hydrops, suggesting that endolymphatic hydrops may be one of the potential key links through which the high-altitude hypobaric hypoxia environment affects auditory conduction function.

## Introduction

High-altitude areas have significantly lower atmospheric pressure and oxygen partial pressure than plain areas due to their high elevation. The unique hypobaric and hypoxic environment in these regions exerts multidimensional impacts on human physiological functions. Specifically, for individuals with rapid first entry into high altitudes, varying degrees of acute mountain sickness often occur, with typical symptoms including dizziness, headache, tinnitus, fatigue, palpitations, loss of appetite, and insomnia [1]; some individuals may also be accompanied by hearing loss [1,2]. In contrast, people who have lived in high-altitude areas for a long time, due to prolonged exposure to this hypobaric and hypoxic environment, may develop chronic mountain sickness (CMS), whose core symptoms also include headache, dizziness, tinnitus, sleep disorders, and fatigue [3].

These studies suggest that the auditory system is highly sensitive to both acute and chronic stimuli of high-altitude hypobaric hypoxia, and may be one of the target organs affected relatively early in this environment [4].

Studies have shown that the inner ear exhibits high sensitivity to changes in barometric pressure and has a high metabolic rate [5,6]. In a hypobaric hypoxia environment, both the auditory and vestibular functions of subjects decrease [7]; animal experiments have also confirmed that a high-altitude hypobaric hypoxia environment can induce hearing loss in rats [1]. However, the specific mechanisms by which the high-altitude hypobaric hypoxia environment affects hearing remain unclear to date. Based on this, the present study aims to use auditory electrophysiological testing indicators, including ABR (Auditory Brainstem Response), ECochG (Electrocochleography), and DPOAE (Distortion Product Oto-acoustic Emissions), to systematically investigate the underlying mechanisms by which this environment impacts auditory function, in order to provide experimental evidence for clarifying the relevant pathophysiological processes.

The auditory brainstem response (ABR) mainly reflects the electrical activity status of the auditory afferent pathway. It has clear waveform characteristics and is primarily composed of 5 typical characteristic waves, with each wave corresponding to a specific neural structure: Wave I originates from the cochlear nerve, Wave II from the cochlear nucleus, Wave III from the superior olivary complex, Wave IV from the lateral lemniscus, and Wave V from the inferior colliculus [8,9]. The function of the auditory conduction pathway can be evaluated through parameters such as the latency and amplitude of each wave.

Electrocochleography (ECochG) is an electrophysiological test that evaluates the functional status of the inner ear (primarily the cochlea) by recording the electrical potentials generated by the cochlea and auditory nerve in response to sound stimulation. It can reflect the activity of cochlear hair cells and the auditory nerve, with its core parameters including the summating potential (SP) and the auditory nerve compound action potential (CAP, also referred to as AP). The SP is a receptor potential generated by cochlear hair cells, essentially representing the DC response of hair cells. It mainly reflects the mechano-electrical transduction properties of hair cells and is formed by the superposition of the positive potential (+SP) produced by outer hair cells (OHC) and the negative potential (-SP) produced by inner hair cells (IHC). In clinical audiometry, when using a click as the stimulation signal, the ECochG waveform recorded at the tympanic membrane (or promontory) is typically a composite wave of SP and AP. Under normal physiological conditions, the positive potential component dominates, and the recorded SP is thus referred to as +SP. If temporary damage to IHC/OHC occurs due to intense noise or acute hypoxia stimulation, the positive potential component decreases while the negative potential component relatively increases, and the SP is then termed -SP. By definition, SP is the algebraic sum of +SP and -SP: when the amplitude of +SP decreases or the amplitude of -SP increases, the SP exhibits an overall negative potential (i.e., -SP). Additionally, SP is characterized by the absence of a refractory period and fatigability. The AP reflects the summation effect of action potentials from thousands of individual auditory nerve fibers, with distinct latency and threshold. It serves as a crucial indicator for evaluating the conduction function of the auditory nerve. In clinical assessment, an amplitude ratio of -SP to AP > 0.4 is generally used as a diagnostic criterion for endolymphatic hydrops, and an area ratio of -SP to AP ≥ 2.0 is determined to be abnormal [9,10].

Otoacoustic emissions (OAE) refer to acoustic energy generated by the active movement of cochlear outer hair cells. This energy is transmitted through the middle ear ossicular chain and tympanic membrane before being released into the external auditory canal. Distortion product otoacoustic emissions (DPOAE) are an important type of OAE. They can only be evoked by simultaneously stimulating the cochlea with two long-duration pure tones (f1, f2) that have a specific frequency ratio and intensity relationship. Their core function is to reflect the functional integrity of inner ear outer hair cells [8,9].

## Materials and methods

### Experimental animals and grouping

Forty healthy male C57BL/6J mice were selected as experimental subjects, aged 6 weeks and weighing 21−27 g, all with no history of exposure to intense noise or use of ototoxic drugs; prior to the experiment, examination via a high-definition

otoscope confirmed that all mice had clean and unobstructed external auditory canals as well as intact and undamaged tympanic membranes. All animal experimental procedures were approved by the Experimental Animal Ethics Committee of Qinghai University Affiliated Hospital (Approval No.: P-SL-2025–330) and strictly complied with the requirements of the Animal Research: Reporting of In Vivo Experiments (ARRIVE Guidelines).

The 40 mice were randomly divided into a control group and an experimental group: The control group was housed in a moderate-altitude environment (altitude: 2260 m). The experimental group was divided into subgroups based on exposure duration to the high-altitude hypobaric hypoxia environment: 3-day, 5-day, 7-day, 10-day, 15-day, 20-day, 25-day, 30-day, and 35-day groups. These mice were housed in a hypobaric hypoxic chamber simulating an environment at 6000 m altitude (temperature: 22 °C, humidity: 55%, air pressure: 46.0 kPa, $CO_2$ concentration: 1776.0 ppm). All mice were purchased from Jiangsu Huachuang Xinnuo Pharmaceutical Technology Co., Ltd., and the experimental procedures were approved by the Ethics Committee of Qinghai University Affiliated Hospital.

## Experimental environment and equipment

All tests were conducted in a shielded room that meets national standards. Equipment used included: Neuro-Audio (Russia) auditory evoked potential instrument (Version 1.0.106.0) and RWD R540i small animal gas anesthetic machine.

## Experimental methods

**Pre-experimental preparation.** Isoflurane inhalation anesthesia was administered at concentrations of 3%−5% for induction and 1%−2% for maintenance. Anesthetized mice were immobilized in a shielded room with their heads maintained in a horizontal and midline position, and the auricles were gently retracted to ensure patency of the external auditory canals. A heating pad was used to maintain the mice's body temperature at 37±0.5 °C, ensuring stable physiological conditions throughout the procedures.

**ABR measurement.** Subcutaneous needle electrodes were used for recording [8,11]: the recording electrode was placed subcutaneously at the intersection of the line connecting the two external auditory meatuses and the midline of the cranium. The reference electrode was placed subcutaneously at the mastoid process of the test ear. The ground electrode was placed subcutaneously at the contralateral hind limb. The stimuli used included click stimuli and 4000 Hz, 8000 Hz tone bursts. Stimulus intensity started from 90 dB SPL and decreased in 10 dB steps, with a cutoff intensity of 20 dB SPL (electrode resistance ≤ 3 Ω). Each stimulus intensity was averaged over 1000 repetitions. The ABR threshold of the test ear was defined as the minimum intensity at which Wave II or Wave III could be repeatedly elicited (Fig 1). For intergroup comparison, the latencies of each wave under 90 dB SPL stimulation were analyzed.

**DPOAE testing.** Under a high-definition otoscope, remove the ABR test microphone and replace it with a DPOAE-dedicated microphone. Ensure the microphone is parallel to the external auditory canal, with the electrode positions remaining unchanged. The device automatically outputs f1 and f2 stimulus tones and simultaneously collects DPOAE signals in the external auditory canal; after the collection of one frequency point is completed, it automatically switches to the next frequency point. Real-time monitoring of the Signal-to-Noise Ratio (SNR) is performed, with a requirement of SNR ≥ 3 dB; if the SNR is too low, it is necessary to check whether the probe has shifted.

**EcochG testing.** A high-definition otoscope combined with a surgical microscope was used to ensure clear exposure of the external auditory canal and tympanic membrane. A disposable needle-shaped recording electrode was accurately inserted into the submucosa of the promontory in the test ear, and stimulation was applied using 125 dB SPL clicks; the reference electrode was implanted in the subcutaneous area of the contralateral mastoid, and the ground electrode was implanted in the subcutaneous area of the contralateral hind limb. First, the baseline electroencephalogram (EEG) was collected; after the fluctuation amplitude of the signal baseline was ≤ 5 μV, the collection of EcochG signals was initiated. The left and right ears were recorded independently, and at least 2 valid repeated recordings were completed for each

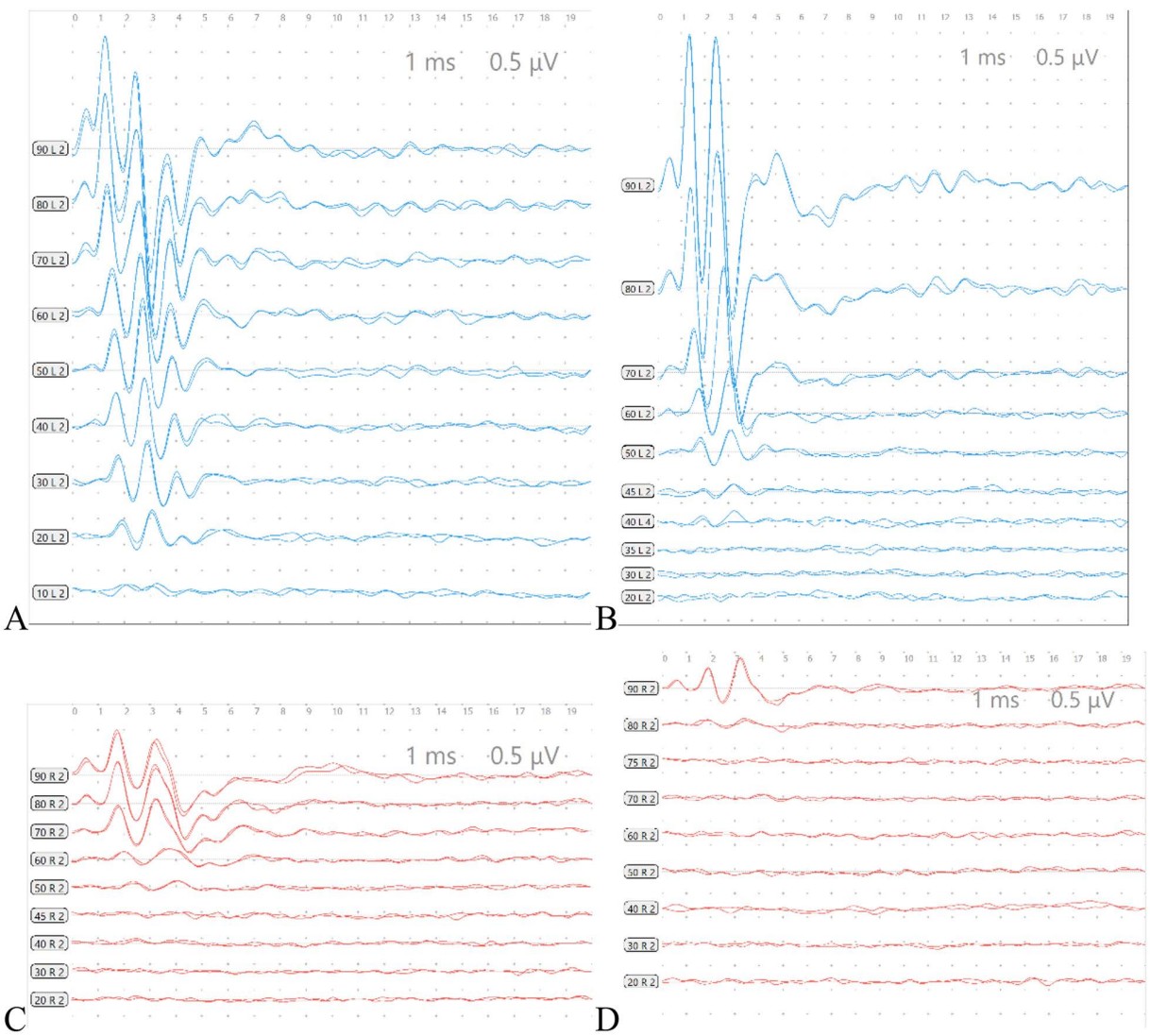

**Fig 1. Labeling of Each Wave of ABR and Determination of Threshold. A:** ABR threshold is 20 dB SPL. **B:** ABR threshold is 40 dB SPL **C:** ABR threshold is 50 dB SPL. **D:** ABR threshold is 80 dB SPL.

ear; the number of stimulus superpositions for a single recording was fixed at 1000, and the stimulus interval was ≥ 10 s to avoid auditory fatigue. The EcochG waveforms of repeated recordings must meet the requirements of amplitude coefficient of variation ≤ 15% and latency difference ≤ 0.2 ms; otherwise, the electrode position must be readjusted before recording again [12–15].

## Statistical methods

Intergroup Comparison: for data that conformed to a normal distribution and showed homogeneity of variance, the independent samples t-test was used. For data that conformed to a normal distribution but had heterogeneous variance, the independent samples t′-test was used. For data that did not conform to a normal distribution, the Mann-Whitney U test (a type of non-parametric test) was used.

Paired Comparison Before and After Reoxygenation: if the data conformed to a normal distribution, the paired samples t-test was used. If the data did not conform to a normal distribution, the Wilcoxon signed-rank test was used.

Statistical analysis was performed using SPSS 31.0 software. All experimental data were expressed as mean ± standard deviation ($\bar{x} \pm s$), and a P-value ≤ 0.05 was considered statistically significant.

## Results

### ABR results

**Click stimulation results. Latency results of each click-evoked wave:** Compared with the control group, under click stimulation, the latency of Wave I began to prolong from Day 3; by Day 15, the degree of latency prolongation reached its peak, and after Day 25, the latency of Wave I gradually returned to the normal level. Among them, the degree of latency prolongation showed a decreasing trend in the 10-day group and 20-day group (Fig 2A).

In the detection of ABR Wave II latency, except for the 35-day group, the latency of Wave II in all other groups was prolonged compared with the control group. Among these groups, the 7-day group had the maximum degree of Wave II latency prolongation, followed by the 10-day group; notably, the latency of Wave II in the 35-day group was not prolonged but was significantly shorter than that in the control group ($P < 0.05$) (Fig 2B).

For ABR Wave III latency, the 7-day group and 10-day group still showed the most significant degree of prolongation; the latency of the 35-day group was shorter than that of the control group, showing a changing trend consistent with that of Wave II latency. In addition, there was no significant difference in the latency of Wave III between the 15-day group, 20-day group and the control group ($P > 0.05$) (Fig 2C).

The latency of ABR Wave IV did not start to prolong until Day 5; among all groups, the 30-day group had the maximum degree of prolongation, followed by the 7-day group; there was no significant difference in the latency of Wave IV between the 20-day group, 25-day group and the control group ($P > 0.05$); while the latency of Wave IV in the 35-day group returned to the normal level of the control group ($P < 0.05$) (Fig 2D).

For ABR Wave V latency, the 7-day group and 10-day group had the most significant degree of prolongation; starting from the 15-day group, the latency of Wave V gradually returned to the normal level of the control group (Fig 2E).

The I-III interwave interval also showed a dynamic change of first prolonging, then decreasing, and eventually becoming lower than that of the control group: among all groups, the 5-day group had the maximum degree of prolongation, followed by the 3-day group and 7-day group; starting from Day 10, the I-III interwave interval of each group gradually decreased and finally returned to the level of the control group; by Day 35, the I-III interwave interval of this group was significantly lower than that of the control group ($P < 0.05$) (Fig 2F).

**Amplitude Results of Click-Evoked Wave II and Wave III:** Amplitude of Wave II: compared with the control group, the amplitude of Wave II increased in the 3-day, 5-day, 15-day, 30-day, and 35-day groups, while it decreased in the 10-day, 20-day, and 25-day groups; the amplitude of Wave II in the 7-day group showed no statistical difference from that in the control group ($P > 0.05$) (Fig 3A).

Amplitude of Wave III: compared with the control group, the amplitude of Wave III increased in the 3-day, 5-day, 7-day, 20-day, and 30-day groups, while it decreased in the 25-day and 35-day groups; the amplitude of Wave III in the 10-day and 15-day groups showed no statistical difference from that in the control group ($P > 0.05$) (Fig 3B).

Comparative Analysis of Dynamic Changes in Amplitudes: additionally, it could be observed that the amplitude changes of both Wave II and Wave III exhibited a sawtooth-like trend with alternating rises and falls. The two waves shared similar fluctuation characteristics in their dynamic change patterns. Further comparison revealed that the change trend of Wave II amplitude was more moderate, while the amplitude of Wave III changed relatively faster (Fig 3C).

**Ratio of Wave II to Wave III Under Click Stimulation:** As shown in Fig 4, under click stimulation, all groups had Wave III as the dominant wave except the 25-day group, which had Wave II as the dominant wave.

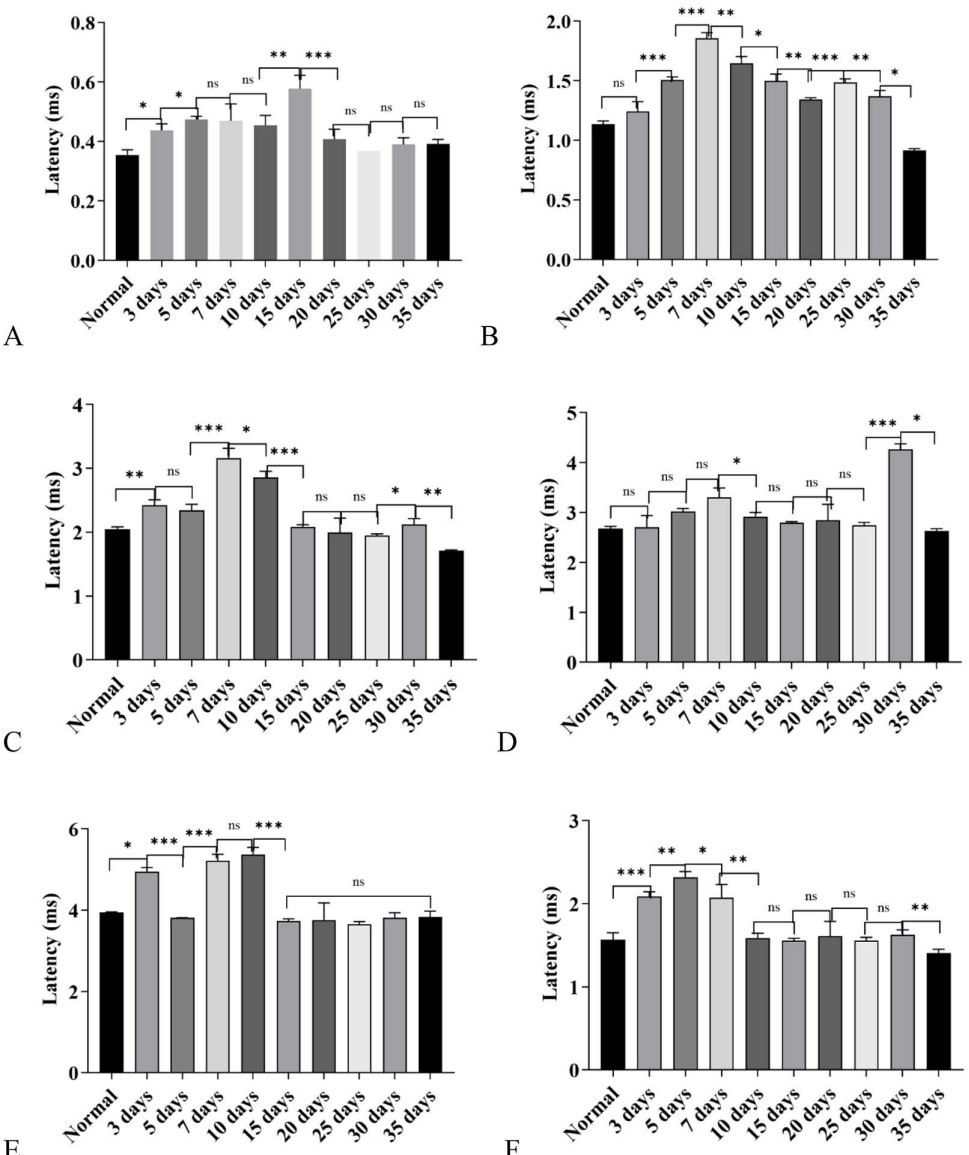

**Fig 2. Changes in Auditory Brainstem Response (ABR) wave latencies in C57BL/6J mice exposed to a high-altitude hypobaric hypoxic environment. A:** Wave I latency: The 15-day group showed the most pronounced prolongation, whereas the 25-, 30-, and 35-day groups exhibited no significant difference compared with controls. **B:** Wave II latency: The 7-day group displayed the greatest prolongation, followed by the 10-day group. In contrast, the 35-day group showed a significantly shorter latency than the control group. **C:** Wave III latency: The 7-day group had the most significant prolongation, followed by the 10-day group. No significant differences were observed between the control, 15-day, and 20-day groups. The 35-day group showed a significantly shorter latency than controls. **D:** Wave IV latency: The 30-day group exhibited the greatest prolongation, followed by the 7-day group. No significant differences were found between the control, 3-day, 20-day, 25-day, and 35-day groups. **E:** Wave V latency: The 7-day and 10-day groups showed the most marked prolongation. No significant differences were observed between the control, 20-day, and 35-day groups, or among the 15- to 35-day groups. **F:** Click I–III interwave interval: The 5-day group exhibited the most pronounced prolongation, followed by the 3-day and 7-day groups. The 10- to 30-day groups showed no difference from controls, while the 35-day group displayed a shorter interval than controls.(ns: $P > 0.05$, *: $P \leq 0.05$, **: $P < 0.01$, ***: $P < 0.001$).

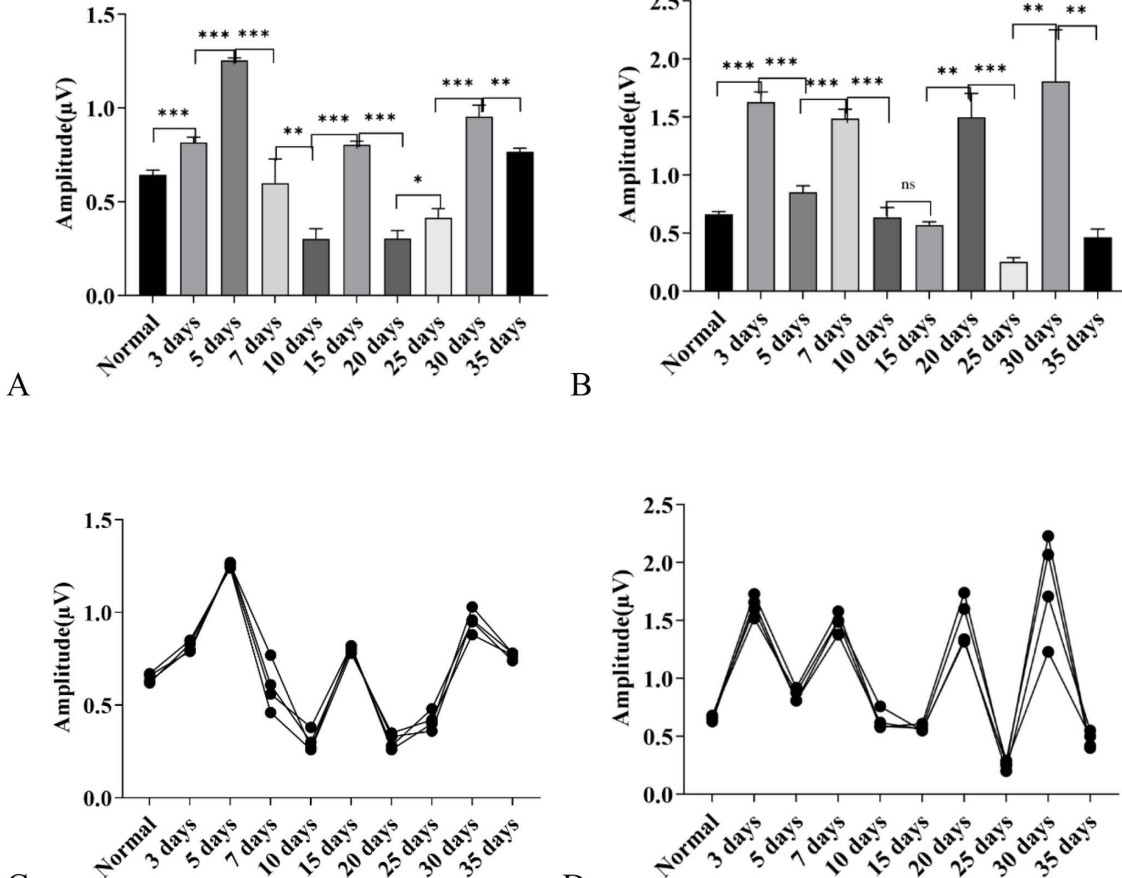

**Fig 3. Changes in Auditory Brainstem Response (ABR) wave amplitudes in C57BL/6J mice exposed to a high-altitude hypobaric hypoxic environment. A:** Wave II amplitude: Significant differences were observed among groups with different exposure durations. The 5-day group showed the greatest increase, followed by the 30-day group. The 10-day and 20-day groups exhibited decreased amplitudes, while the 7-day group showed no significant difference from the control group. **B:** Wave III amplitude: Marked group differences were observed. The 3-, 5-, 7-, 20-, and 30-day groups showed increased amplitudes, whereas the 25- and 35-day groups showed decreases. The 10- and 15-day groups did not differ significantly from the control group. **C:** Overall trend of Wave II amplitude across exposure durations. **D:** Overall trend of Wave III amplitude across exposure durations.(ns: $P > 0.05$, *: $P \leq 0.05$, **: $P < 0.01$, ***: $P < 0.001$).

**Results of 4000 Hz tone burst stimulation. Latency results of each wave under 4000 Hz tone burst stimulation:** Under exposure to the high-altitude hypobaric and hypoxic environment, the latency of ABR Wave I began to prolong from Day 3 of exposure; among all groups, the 30-day group and 7-day group exhibited the most significant prolongation of Wave I latency. Further observation showed that not only did the latency of Wave I in the 35-day group not prolong, but it was also significantly shorter than that in the control group ($P < 0.001$) (Fig 5A).

For Wave II latency, the 7-day group had the most obvious prolongation, followed by the 30-day group; the latency of the 35-day group was shorter than that in the control group ($P < 0.001$) (Fig 5B).

For Wave III latency, the 7-day group had the most obvious prolongation, followed by the 5-day group; similarly, the latency of the 35-day group was shorter than that in the control group ($P < 0.05$) (Fig 5C).

For Wave IV latency, the 7-day group had the most obvious prolongation, followed by the 10-day group; starting from Day 25, the latency began to decrease back to the level of the control group (Fig 5D).

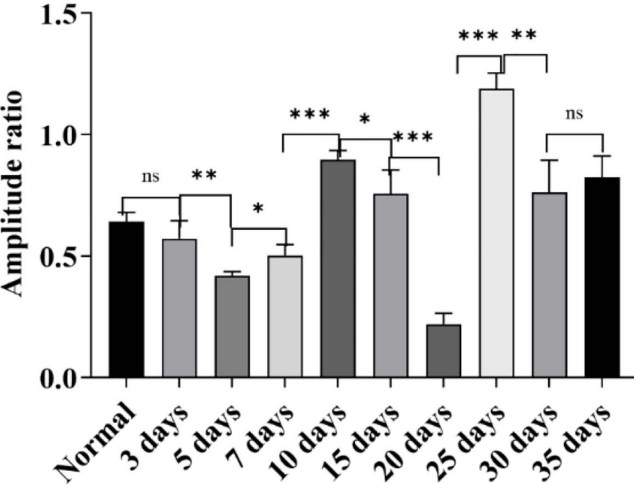

**Fig 4. Shows the characteristics of the amplitude ratio between ABR Wave II and Wave III in each group.** As shown in the figure, the dominant wave type of ABR in each group can be clearly determined through this amplitude ratio: except for the 25-day group, all other groups take Wave III as the dominant wave; only the 25-day group takes Wave II as the dominant wave (ns: $P>0.05$, *: $P\leq0.05$, **: $P<0.01$, ***: $P<0.001$).

In terms of Wave V latency, the 7-day group and 10-day group exhibited the most significant prolongation, and the 35-day group returned to the level of the control group ($P>0.05$) (Fig 5E).

For the I-III interwave interval, the 7-day group had the most obvious prolongation ($P<0.05$), and the 35-day group showed no statistical difference compared with the control group ($P>0.05$) (Fig 5F).

**Amplitude Results of Wave II and Wave III Under 4000 Hz Tone Burst Stimulation:** Results of intergroup comparison of ABR Wave II amplitude showed that: compared with the control group, only the 30-day group exhibited an increase in amplitude; the 7-day group and 25-day group had no statistical difference in Wave II amplitude from the control group ($P>0.05$); the Wave II amplitude of all other groups was lower than that in the control group, and the difference was statistically significant ($P<0.05$) (Fig 6A). Further observation of results under different stimulation conditions revealed that the change trend of Wave II amplitude in the 7-day group under 4000 Hz tone burst stimulation was highly consistent with its change characteristics under click stimulation-both showed no statistical difference from their respective control groups ($P>0.05$). Specifically, under the two stimulation conditions (click and 4000 Hz tone burst), the 7-day group exhibited no significant fluctuation in Wave II amplitude, but showed the most obvious changes in ABR latency. This result suggests that in the high-altitude hypobaric and hypoxic environment exposure model, changes in ABR latency may be more stable and sensitive than changes in amplitude.

For intergroup comparison of Wave III amplitude: the 35-day group showed the most obvious increase in amplitude, followed by the 10-day group and 3-day group; while the 5-day group, 7-day group, 15-day group, 20-day group, 25-day group, and 30-day group exhibited a decrease in Wave III amplitude ($P<0.05$) (Fig 6B).

**Amplitude ratio results of wave II and wave III under 4000 Hz tone burst stimulation:** As shown in Fig 7, under the condition of 4000 Hz tone burst stimulation, the composition of the dominant waves of ABR exhibited a specific pattern: among which, the 25-day group and 35-day group took Wave II as the dominant wave, while all other groups took Wave III as the dominant wave; the distribution characteristics of these dominant waves showed significant differences from the performance of ABR dominant waves under click stimulation.

**Results of 8000 Hz short pure tone stimulation. Latency results of each wave under 8000 Hz short pure tone stimulation:** Under the condition of exposure to the high-altitude hypobaric and hypoxic environment, the latencies of each Auditory Brainstem Response (ABR) wave exhibited different variation characteristics:

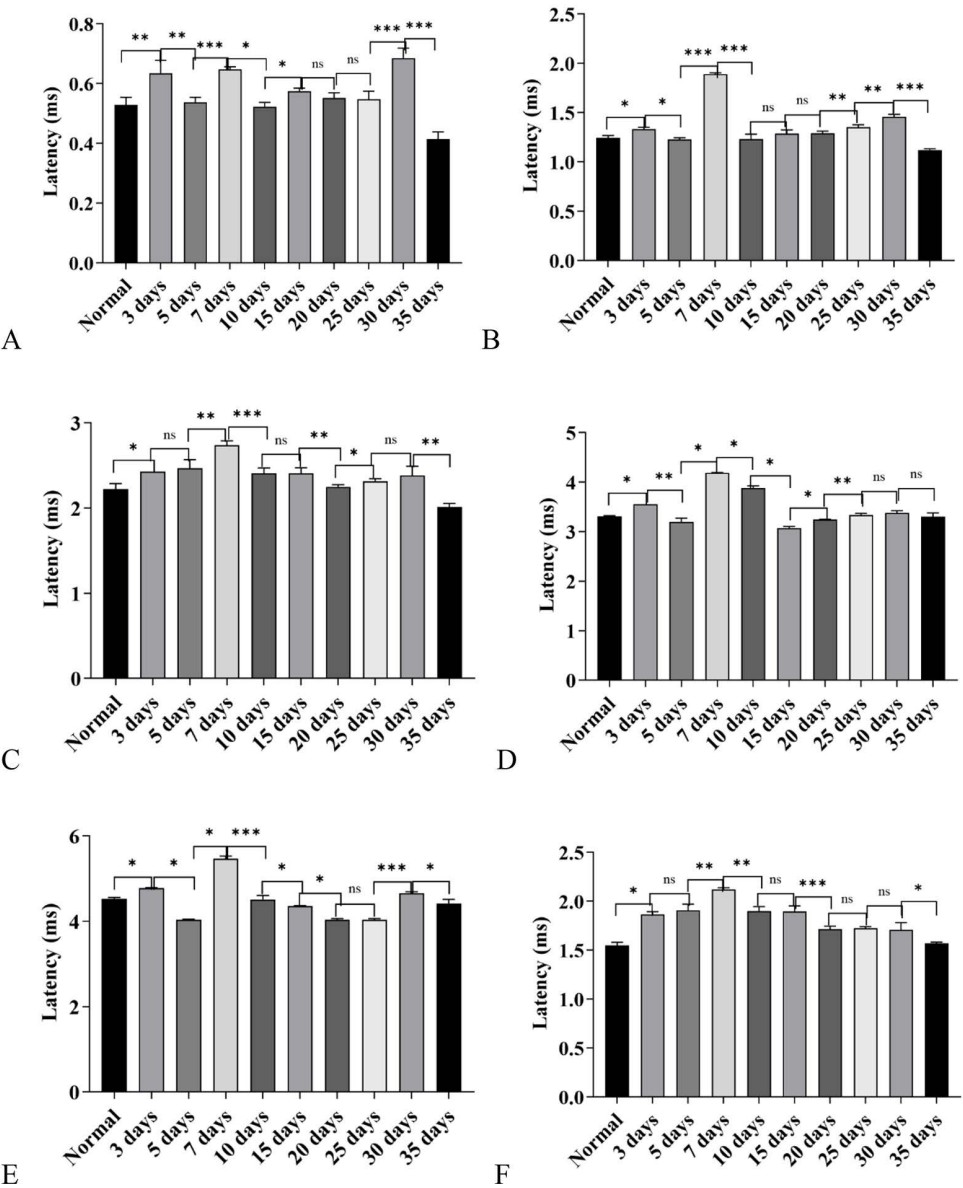

**Fig 5. Distribution of Auditory Brainstem Response (ABR) wave latencies under 4000 Hz tone-burst stimulation in C57BL/6J mice exposed to a high-altitude hypobaric hypoxic environment. A:** Wave I latency: The 3-, 7-, and 30-day groups showed the most pronounced prolongation. The 5-, 10-, 20-, and 25-day groups showed no significant differences from controls, while the 15- and 35-day groups exhibited shorter latencies. **B:** Wave II latency: The 7-day group displayed the most significant prolongation, followed by the 30-day group. The 35-day group showed a significantly shorter latency than controls, while the 5- and 10-day groups showed no difference. **C:** Wave III latency: The 7-day group exhibited the greatest prolongation, followed by the 3- and 5-day groups. The 20- and 25-day groups showed no difference from controls, whereas the 35-day group had a significantly shorter latency. **D:** Wave IV latency: The 7-day group showed the most marked prolongation, followed by the 10-day group. The 25- and 35-day groups showed no significant difference from controls. **E:** Wave V latency: The 3- and 7-day groups showed evident prolongation, while the 10- and 35-day groups did not differ from controls. **F:** I–III interwave interval: All groups except the 35-day group exhibited prolonged intervals; the 7-day group showed the most significant prolongation (ns: $P > 0.05$; *: $P \leq 0.05$; **: $P < 0.01$; ***: $P < 0.001$).

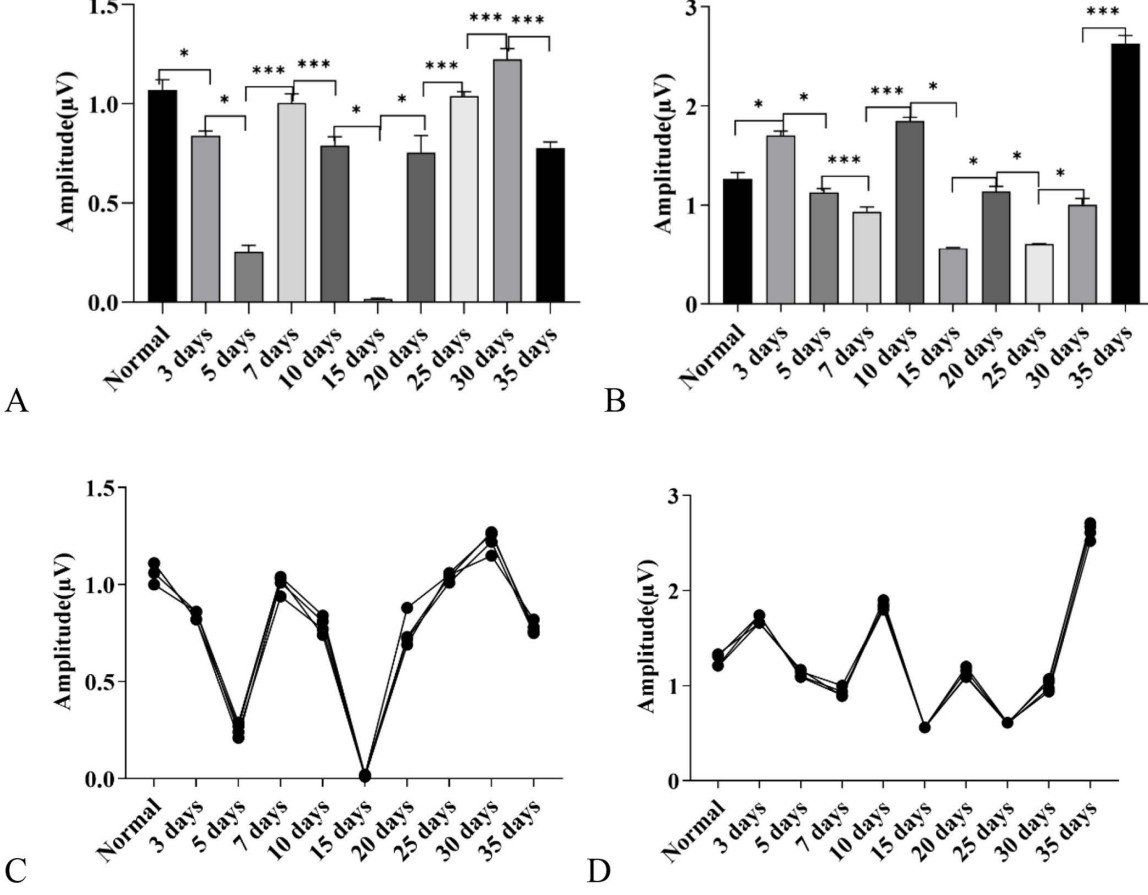

**Fig 6. Distribution of ABR Wave II and Wave III Amplitudes Under 4000 Hz Tone Burst Stimulation. A:** Distribution of ABR Wave II amplitude across groups. As shown in the figure, the Wave II amplitude of the 5-day group and 15-day group was significantly decreased compared with the control group; the 7-day group and 25-day group showed no statistical difference in Wave II amplitude from the control group; the 30-day group was the only group with a higher Wave II amplitude than the control group, and the difference was statistically significant. **B:** Distribution of ABR Wave III amplitude across groups. As seen with the control group as a reference: the 3-day group, 10-day group, and 35-day group all showed an increasing trend in Wave III amplitude; while the 15-day group and 25-day group had the lowest Wave III amplitude compared with all other groups, making them the two groups with the lowest Wave III amplitude levels among all groups. **C:** Figure of the overall change trend of ABR Wave II amplitude in each group. **D:** Figure of the overall change trend of ABR Wave III amplitude in each group (ns: $P > 0.05$, *: $P \le 0.05$, **: $P < 0.01$, ***: $P < 0.001$).

The latency of Wave I was prolonged in all groups, with the most significant prolongation observed in the 20-day group and the 35-day group (Fig 8A).

For the latency of Wave II, it was prolonged in all groups except the 10-day group and the 35-day group. Among these groups, the prolongation effect was most prominent in the 7-day group, 20-day group, and 25-day group. Meanwhile, the latency of Wave II in the 35-day group had returned to the level of the control group (Fig 8B).

The prolongation of Wave III latency was most significant in the 7-day group and the 20-day group, and the latency of Wave III in the 35-day group was significantly shorter than that of the control group ($P < 0.01$) (Fig 8C).

The prolongation of Wave IV latency was also most obvious in the 7-day group and the 20-day group, and the latency of Wave IV in the 35-day group was also shorter than that of the control group ($P < 0.05$) (Fig 8D).

The main groups with prolonged Wave V latency were the 7-day group and the 20-day group, and the latency of Wave V in the 35-day group was still shorter than that of the control group ($P < 0.05$) (Fig 8E).

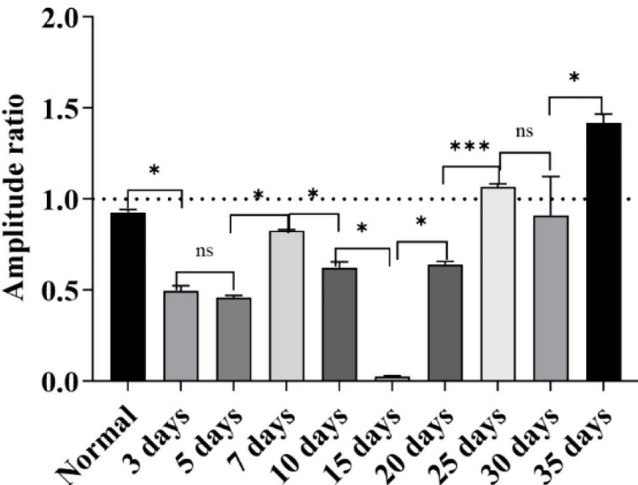

**Fig 7. Figure shows the distribution of the ABR Wave II/Wave III amplitude ratio in each group.** It can be seen that most experimental groups, like the control group, take Wave III as the dominant wave, while the 25-day group and 35-day group take Wave II as the dominant wave (ns: $P > 0.05$, *: $P \leq 0.05$, **: $P < 0.01$, ***: $P < 0.001$).

In the Analysis of ABR-Related Interwave Intervals: for the I-III interwave interval, the prolongation was most significant in the 7-day group and the 5-day group. In contrast, the I-III interwave intervals of the 30-day group and the 35-day group decreased to below the level of the control group, and the difference was statistically significant ($P < 0.05$) (Fig 8F). For the I-V interwave interval, the prolongation was most prominent in the 7-day group and the 20-day group. The I-V interwave interval of the 35-day group showed an adaptive change and decreased to below the level of the control group, with a statistically significant difference ($P < 0.05$) (Fig 8G). For the III-V interwave interval, the prolongation was most obvious in the 7-day group and the 20-day group. Similarly, the III-V interwave interval of the 35-day group was shorter than that of the control group, and the difference was statistically significant ($P < 0.05$) (Fig 8H).

**Results of wave II and wave III amplitudes under 8000 Hz short pure tone stimulation:** In the analysis of Auditory Brainstem Response (ABR) wave amplitudes: the amplitude of Wave II showed differentiated changes: the 5-day group and 15-day group exhibited an increasing trend, while the 3-day group, 7-day group, 10-day group, 20-day group, 25-day group, and 35-day group showed a decreasing trend. The amplitude of Wave II in the 30-day group had no significant difference compared with the control group (Fig 9A). Regarding the amplitude of Wave III, the increase was most significant in the 10-day group. The amplitude of Wave III in the 30-day group decreased to the level of the control group (i.e., normal level), and the amplitudes of Wave III in all other groups were lower than that of the control group (Fig 9B).

**Results of the amplitude ratio between wave II and wave III:** Under the high-altitude hypobaric and hypoxic environment, among all ABR waves induced by 8000 Hz short pure tone stimulation, Wave II was consistently the dominant wave (Fig 10).

**Inter-group comparison among different stimulus sound groups.** Based on the above-mentioned Auditory Brainstem Response (ABR) latency detection results, to conduct an in-depth analysis of the response characteristics of each group to the high-altitude hypobaric and hypoxic environment under different stimulus sounds, the 7-day group-which exhibited the most significant response to this environment-was selected for inter-group comparative analysis. Further research revealed that, compared with other ABR waves, the latency of Wave II exhibited the most prominent response to the high-altitude hypobaric and hypoxic environment, and this response characteristic was more pronounced in the low-frequency stimulus groups. Furthermore, in the 8000 Hz stimulus group, the response effects of all ABR waves

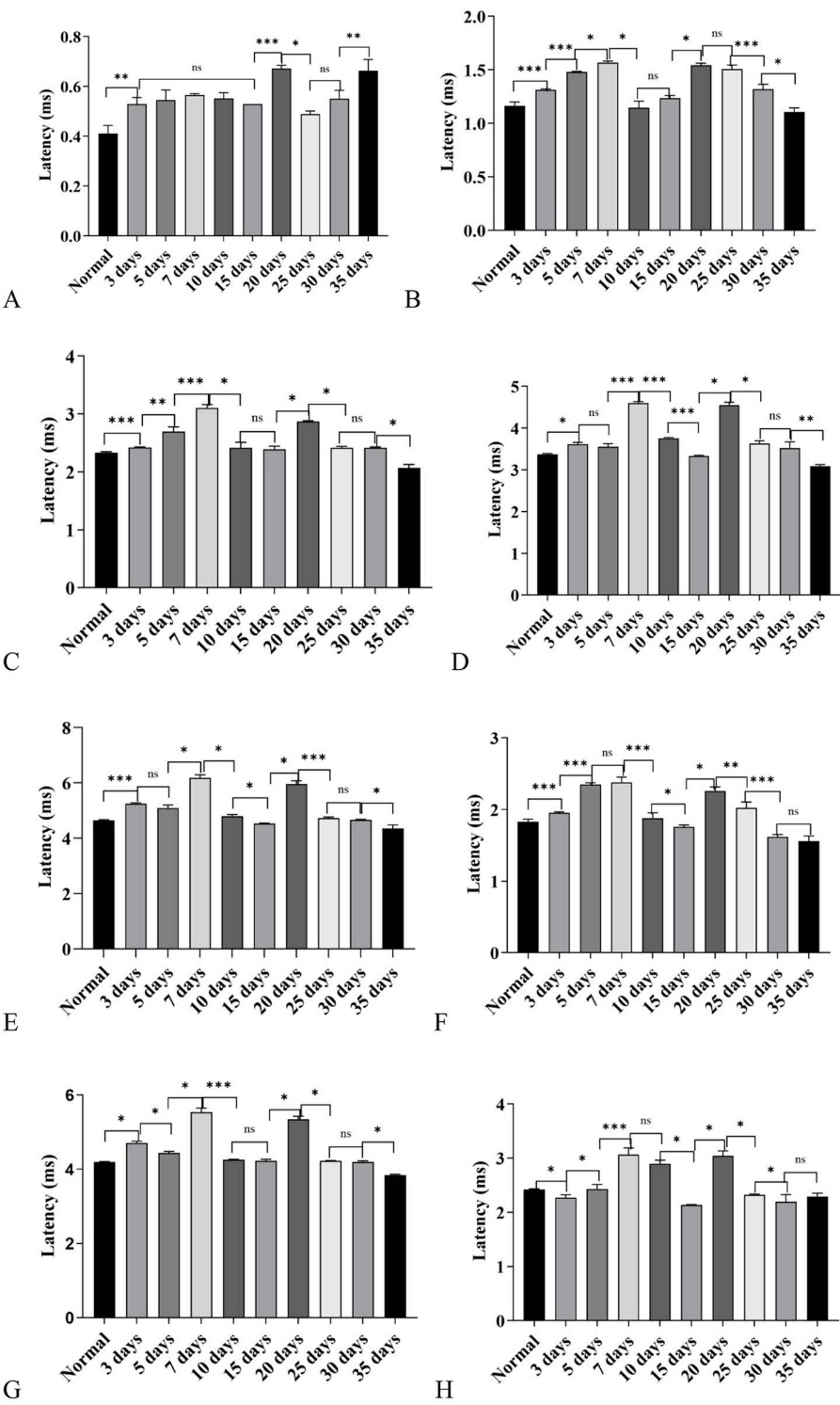

**Fig 8. Distribution of Auditory Brainstem Response (ABR) wave latencies and interwave intervals under 8000 Hz tone-burst stimulation in C57BL/6J mice exposed to a high-altitude hypobaric hypoxic environment. A:** Wave I latency: All exposure groups showed prolonged latencies compared with controls ($P<0.05$), with the 20- and 35-day groups exhibiting the greatest prolongation. **B:** Wave II latency: The 7-day and 20-day groups showed the most significant prolongation. No significant differences were found in the 10-day and 35-day groups ($P>0.05$). **C:** Wave III latency: The 7-day group showed the most pronounced prolongation. The 10- and 15-day groups did not differ from controls ($P>0.05$), while the 35-day group exhibited a significantly shorter latency ($P<0.01$). **D:** Wave IV latency: The 7-day and 20-day groups showed marked prolongation. The 15- and 30-day

groups showed no significant difference (*P*>0.05), whereas the 35-day group exhibited shorter latency (*P*<0.05). **E:** Wave V latency: The 7-day and 20-day groups showed the greatest prolongation. The 30-day group showed no difference from controls (*P*>0.05). The 15- and 35-day groups exhibited significantly shorter latencies (*P*<0.05). **F:** I-III interwave interval: The 5-day and 7-day groups showed the most significant prolongation, while the 10-day group showed no difference (*P*>0.05). The 15-, 30-, and 35-day groups exhibited shorter intervals than controls (*P*<0.05). **G:** I-V interwave interval: Prolongation was most prominent in the 7-day and 20-day groups. The 35-day group showed a shorter interval (*P*<0.05), while the 15-, 25-, and 30-day groups did not differ significantly from controls (*P*>0.05). **H:** III-V interwave interval: The 7-day and 20-day groups showed significant prolongation. The 15- and 35-day groups exhibited shorter intervals (*P*<0.05), and the 5-day group showed no difference from controls (*P*>0.05).(ns: *P*>0.05, *: *P*≤0.05, **: *P*<0.01, ***: *P*<0.001).

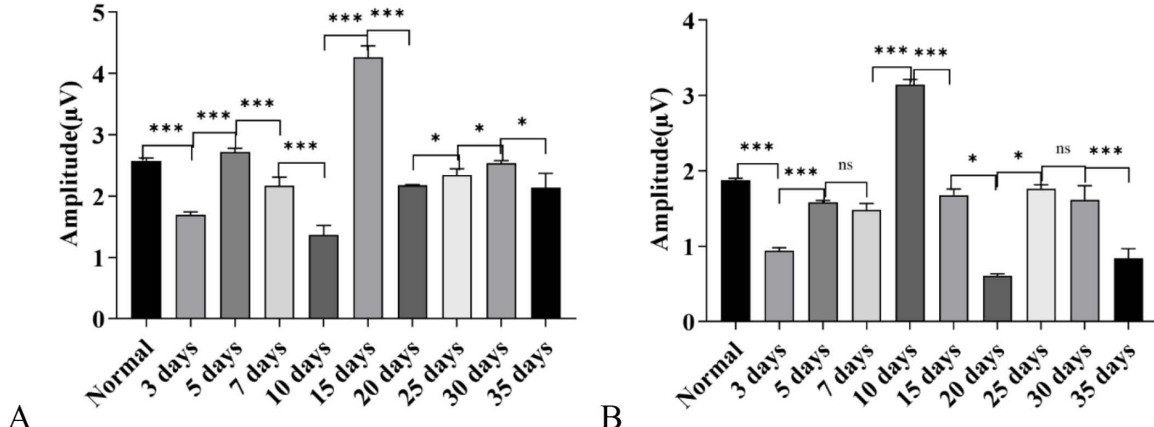

**Fig 9. Changes in Amplitudes of ABR Wave II and Wave III under 8000 Hz Short Pure Tone Stimulation. A:** This panel shows the distribution of ABR Wave II amplitudes in each group. Among them, the 15-day group had the highest Wave II amplitude, the 10-day group had the lowest, and the 30-day group showed no significant statistical difference compared with the control group (*P*>0.05). **B:** This panel shows the distribution of ABR Wave III amplitudes in each group. Among them, the 10-day group had the highest Wave III amplitude, the 20-day group had the lowest, and the 30-day group showed no significant statistical difference compared with the control group (*P*>0.05) (ns: *P*>0.05, *: *P*≤0.05, **: *P*<0.01, ***: *P*<0.001).

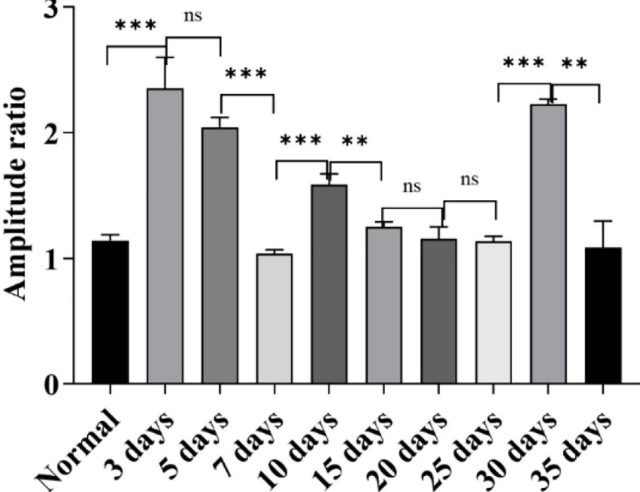

**Fig 10. Figure depicts the distribution of ABR Wave II/III amplitude ratios in each group.** Notably, Wave II was identified as the dominant wave across all groups. Specifically, the Wave II/III amplitude ratios of the 20-day, 25-day, and 35-day groups exhibited no statistically significant differences relative to the control group (*P*>0.05) (ns: *P*>0.05, *: *P*≤0.05, **: *P*<0.01, ***: *P*<0.001).

to the high-altitude hypobaric and hypoxic environment tended to be similar, with no significant interwave differences observed (Fig 11).

**ABR threshold results. Click threshold detection results:** Except for the 5-day and 10-day post-exposure groups, whose auditory thresholds showed no significant difference from those of the control group, mice in all other exposure groups exhibited a clear elevation in auditory thresholds following high-altitude hypobaric and hypoxic exposure. The thresholds began to rise on the third day of exposure and reached their peak levels on days 15 and 25.

Notably, the average click threshold of the control group was 39 dB SPL, which is higher than the well-established normal auditory brainstem response (ABR) threshold of 30 dB SPL typically observed at sea level [16]. This discrepancy can be attributed to the fact that ABR measurements for the control group were conducted at a mid-altitude location (2250 m above sea level). Such an environmental factor may introduce a mild elevation in measured thresholds but does not compromise the validity of the comparative analysis across experimental groups (Fig 12A).

**4000 Hz short pure tone threshold detection results:** Except for the 10-day group, whose threshold showed no difference from that of the control group, all other groups exhibited an increasing trend in threshold. Among these groups, the 15-day, 20-day, 25-day, and 30-day groups had the most significant threshold increase, followed by the 3-day, 5-day, and 7-day groups; in contrast, the 35-day group exhibited a declining trend (Fig 12B).

**8000 Hz short pure tone threshold detection results:** Except for the 10-day group, whose threshold showed no difference from that of the control group, all other groups exhibited an increasing trend in threshold. Among these groups, the 20-day, 25-day, and 30-day groups had the most significant threshold increase, followed by the 3-day, 5-day, and 7-day groups; however, although the threshold of the 35-day group exhibited an obvious declining trend, it was still higher than that of the control group (Fig 12C).

**Comparative analysis of inter-group detection results for ABR thresholds among the above three groups:** Inter-group comparison of ABR thresholds under different stimulus conditions revealed the following: after 15–20 days of exposure to the high-altitude hypobaric and hypoxic environment, the ABR thresholds reached their peak increase; after 35 days of exposure, the ABR thresholds exhibited a declining trend. Notably, the 10th day was a critical time point-specifically, the ABR thresholds under click, 4000 Hz short pure tone, and 8000 Hz short pure tone stimulation all decreased back to the level of the normal control group on the 10th day of exposure.

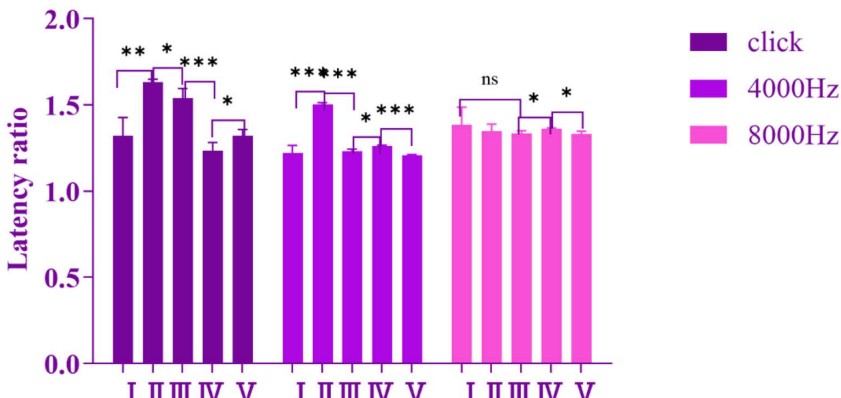

**Fig 11. Shown in the figure is the distribution of the ratios of the latencies of each ABR wave to those of their respective control groups under different stimulus frequencies (click, 4000 Hz, and 8000 Hz).** Under the click stimulus condition, the latency prolongation ratios of Wave II and Wave III were the highest. Under the 4000 Hz stimulus condition, Wave II exhibited the highest latency prolongation ratio. Whereas under the 8000 Hz stimulus condition, the latency prolongation ratios among all waves were negligible, with no significant differences observed (ns: $P > 0.05$, *: $P \leq 0.05$, **: $P < 0.01$, ***: $P < 0.001$).

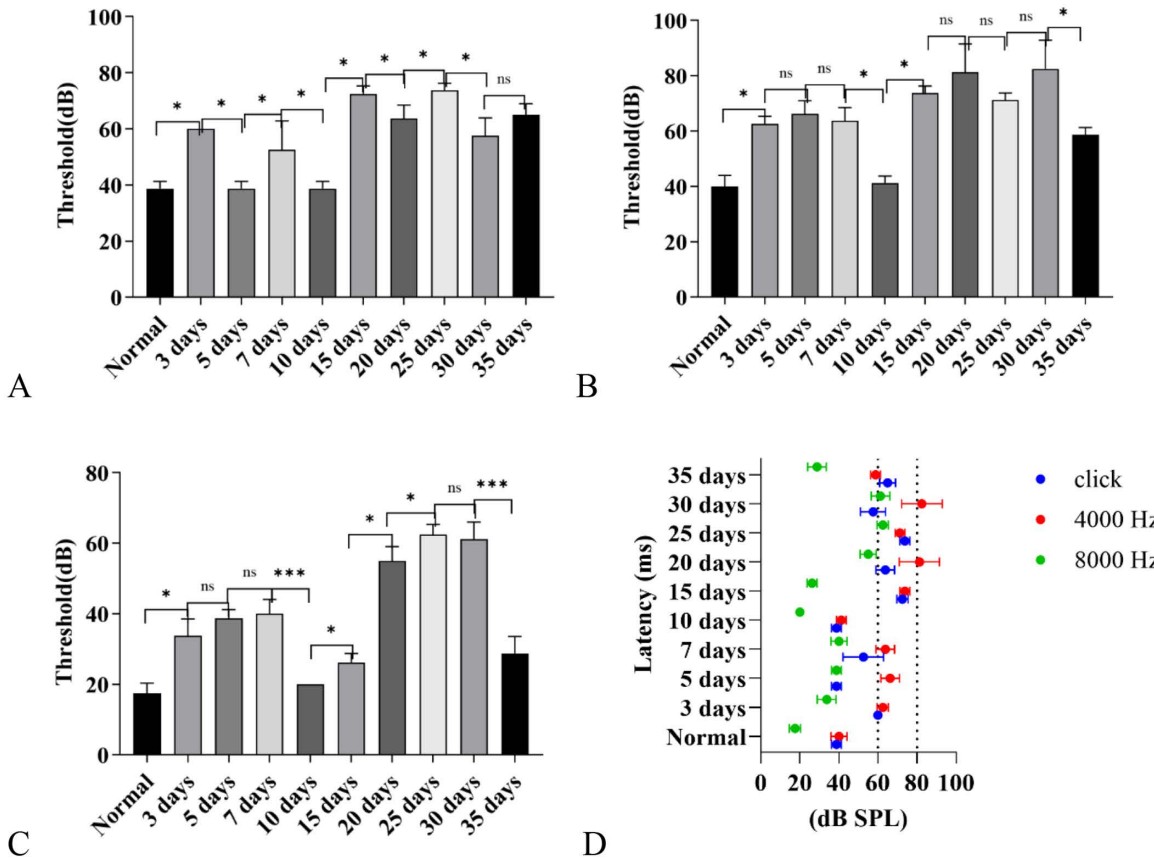

**Fig 12. Distribution of Auditory Brainstem Response (ABR) thresholds at different stimulus frequencies in C57BL/6J mice exposed to a high-altitude hypobaric hypoxic environment. A:** Click stimulation: The 15- and 25-day groups showed the most pronounced threshold elevation, whereas the 5- and 10-day groups showed no significant difference from controls ($P > 0.05$). **B:** 4000 Hz tone stimulation: All groups except the 10-day group exhibited elevated thresholds. Although the 35-day group showed a slight decline, its threshold remained significantly higher than that of the control group ($P < 0.05$). **C:** 8000 Hz tone stimulation: The 25- and 30-day groups showed the greatest threshold increase. The 10-day group showed no difference from controls ($P > 0.05$). The 35-day group showed a mild decline but remained significantly higher than the control group ($P < 0.05$). **D:** Summary: Within the first 10 days of exposure, hearing loss reached approximately 60 dB under click and 4000 Hz stimulation, and 40 dB under 8000 Hz stimulation. After 10 days, thresholds further increased-up to 70 dB for click, ~80 dB for 4000 Hz, and ~60 dB for 8000 Hz stimuli. (ns: $P > 0.05$, *: $P \leq 0.05$, **: $P < 0.01$, ***: $P < 0.001$).

Furthermore, additional comparative analysis showed that, compared with the 8000 Hz short pure tone stimulus group, the ABR thresholds of the click stimulus group (2000–4000 Hz) and the 4000 Hz short pure tone group increased more significantly under exposure to the high-altitude hypobaric and hypoxic environment. This result suggests that the high-altitude hypobaric and hypoxic environment has significant frequency selectivity in its impact on auditory function, and exhibits a stronger tendency to impair low-frequency hearing (Fig 12D).

**Comparative analysis of ABR threshold and ABR latency results:** In the high-altitude hypobaric and hypoxic environment, the latency response of the 7-day group was both significant and stable. However, while the ABR thresholds of the 7-day group all showed a significant increase, the magnitude of this increase did not exhibit the same level of significance as the latency response. It is hypothesized that this phenomenon is associated with the negative feedback mechanism present in the auditory electrical signal conduction pathway, and this mechanism can partially inhibit the hearing loss effect induced by the accumulation of prolonged latency.

**ABR results summary.** **Summary of ABR latency results:** Under the stimulation of click (tone burst), 4000 Hz short pure tone, and 8000 Hz short pure tone, among the groups with different exposure durations, the latency response of the 7-day group to the high-altitude hypobaric and hypoxic environment was the most stable and significant. Secondly, under click and 4000 Hz stimulation, the magnitude of the latency response of the 10-day group was second only to that of the 7-day group; whereas under 8000 Hz stimulation, the group with the second most significant latency response after the 7-day group was the 20-day group.

Among each ABR wave in all groups, the latency of Wave II (cochlear nucleus) exhibited the most obvious response to exposure to the high-altitude hypobaric and hypoxic environment, and this response characteristic was particularly prominent under low-frequency stimulation conditions.

In the high-altitude hypobaric and hypoxic environment, the ABR of C57BL/6J mice may cope with environmental changes through a fluctuating adaptive regulation mechanism, and the adaptive regulation process of mice to this environment is typically completed on the 35th day of exposure.

**Summary of ABR threshold results:** On the 3rd day of exposure to the high-altitude hypobaric and hypoxic environment, the ABR thresholds of C57BL/6J mice began to increase; by the 10th day of exposure, their ABR thresholds had all decreased back to the normal level, followed by another increase. Starting from the 35th day of prolonged exposure, the ABR thresholds of the mice exhibited a declining trend once again.

During the exposure to the high-altitude hypobaric and hypoxic environment, C57BL/6J mice showed an obvious low-frequency bias in hearing loss, and the magnitude of this low-frequency hearing loss typically reached its peak between the 15th and 20th days of exposure.

For the 7-day exposure group, the distribution of its ABR thresholds did not exhibit the significant and stable characteristics to the same extent as those of the latencies. This phenomenon is hypothesized to be associated with the negative feedback mechanism present in the auditory electrical signal conduction pathway; through its regulatory effect, this mechanism can partially inhibit the hearing loss effect induced by the accumulation of prolonged latencies.

## EcochG results

**SP latency results.** Under exposure to the high-altitude hypobaric and hypoxic environment, the changes in -SP latency across each exposure group were as follows: except for the 15-day exposure group, which showed no significant difference from the control group, all other groups exhibited prolongation in -SP latency. Among these groups, the 25-day, 30-day, and 7-day exposure groups had the most significant prolongation in latency, while the 35-day exposure group exhibited a declining trend in -SP latency (Fig 13A).

For each group of C57BL/6J mice, when comparing the -SP latencies before and after hypobaric hypoxia (HBH) and repressurization-reoxygenation (R-R), it was observed that the -SP latencies after R-R were all shorter than those under HBH (Fig 13B). Specifically, the 3-day, 10-day, 15-day, and 20-day groups showed significant recovery effects in -SP latency after R-R; in contrast, the 7-day, 25-day, and 30-day groups had the most obvious prolongation in -SP latency and the poorest recovery effects after reoxygenation. Among them, the relative recovery rates of latency after reoxygenation in the 15-day group and 3-day group reached 46.3% and 45.8%, respectively (Fig 13C).

**AP latency results.** Under exposure to the high-altitude hypobaric and hypoxic environment, the changes in AP (Compound Action Potential) latency across each group were as follows: Except for the 15-day group and 35-day group, whose AP latencies showed no significant difference from that of the control group, all other groups exhibited a significant prolongation in AP latency. Among these groups, the 7-day group and 10-day group had the most obvious prolongation effect on AP latency (Fig 14A).

When comparing the AP latencies of each group before and after hypobaric hypoxia (HBH) and repressurization-reoxygenation (R-R), the following was observed: After R-R, the recovery effects of AP latency varied among groups (Fig 14B). Specifically, the 5-day, 15-day, and 35-day groups showed poor recovery effects after R-R, while the 10-day

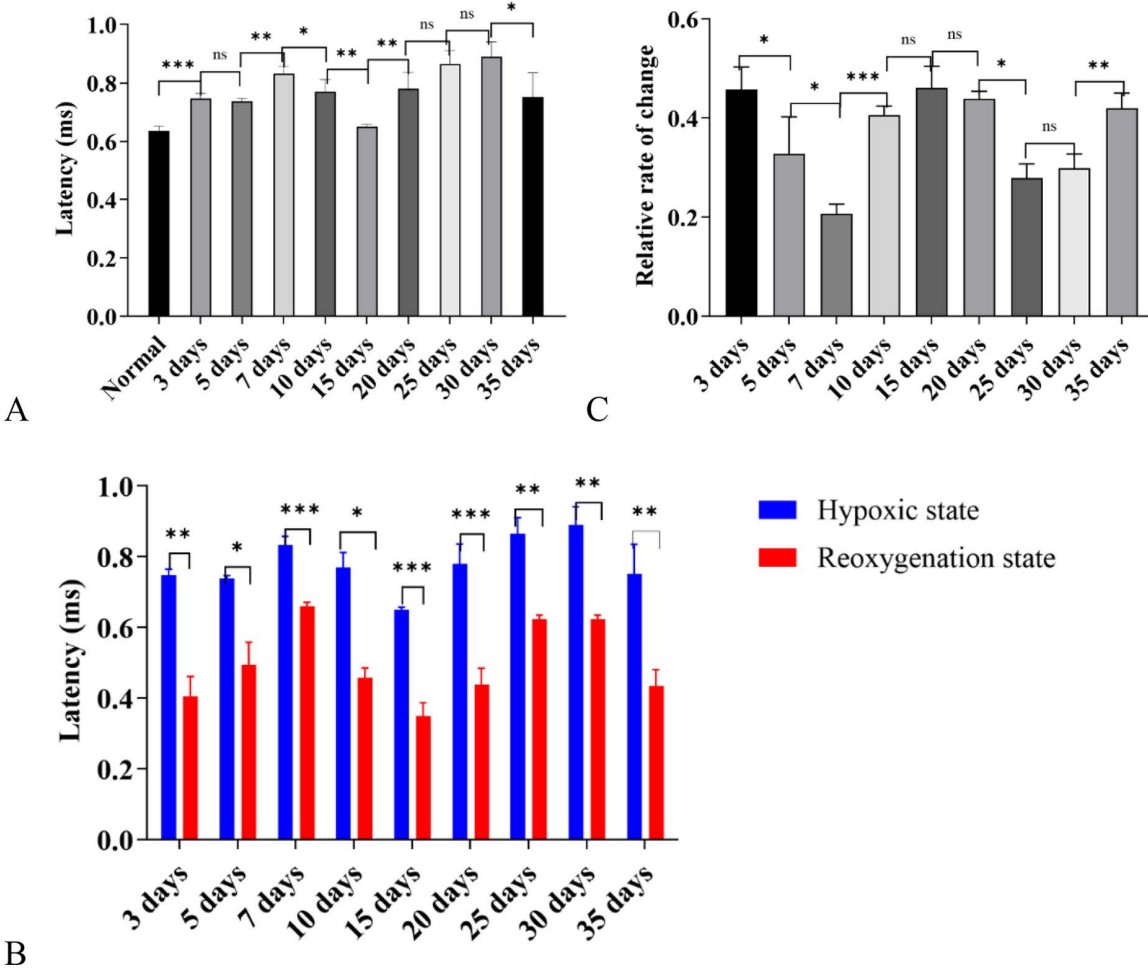

**Fig 13. Distribution of Electrocochleography (EcochG) negative summating potential (-SP) in C57BL/6J mice before and after exposure to a high-altitude hypobaric hypoxic (HBH) environment. A:** -SP latency across groups: The 15-day group showed no significant difference from controls ($P > 0.05$). All other groups exhibited prolonged -SP latencies, with the 7-, 25-, and 30-day groups showing the most pronounced prolongation. **B:** -SP latency before and after repressurization–reoxygenation **(R-R):** Following R-R, all groups showed a general decline in -SP latency compared with the HBH condition. **C:** Ratio of recovery in -SP latency (Δ-SP latency/pre-R–R latency): The 3-, 10-, 15-, and 20-day groups demonstrated better recovery after R–R, while the 7-, 25-, and 30-day groups exhibited the most pronounced prolongation under HBH and the poorest recovery following R-R. (ns: $P > 0.05$, *: $P \le 0.05$, **: $P < 0.01$, ***: $P < 0.001$).

group and 20-day group had the best recovery effects. Further data revealed that after R-R, the relative recovery rates of AP latency in the 10-day group and 20-day group were 34.8% and 29.5% (Fig 14C), respectively-significantly lower than those of the negative Summating Potential (-SP).

**Comparative analysis of -SP and AP latency results.** In the high-altitude hypobaric and hypoxic environment, the 7-day group still exhibited relatively stable and significantly responsive characteristics in terms of both -SP latency and AP latency.

By comparing the recovery efficiency of the two (i.e., -SP and AP) before and after repressurization-reoxygenation (R-R), it can be observed that the sensitivity of -SP latency to the hypobaric hypoxic environment was significantly higher than that of AP latency. This finding suggests that the cochlea (inner hair cells) is more sensitive to hypobaric hypoxia than auditory nerve fibers.

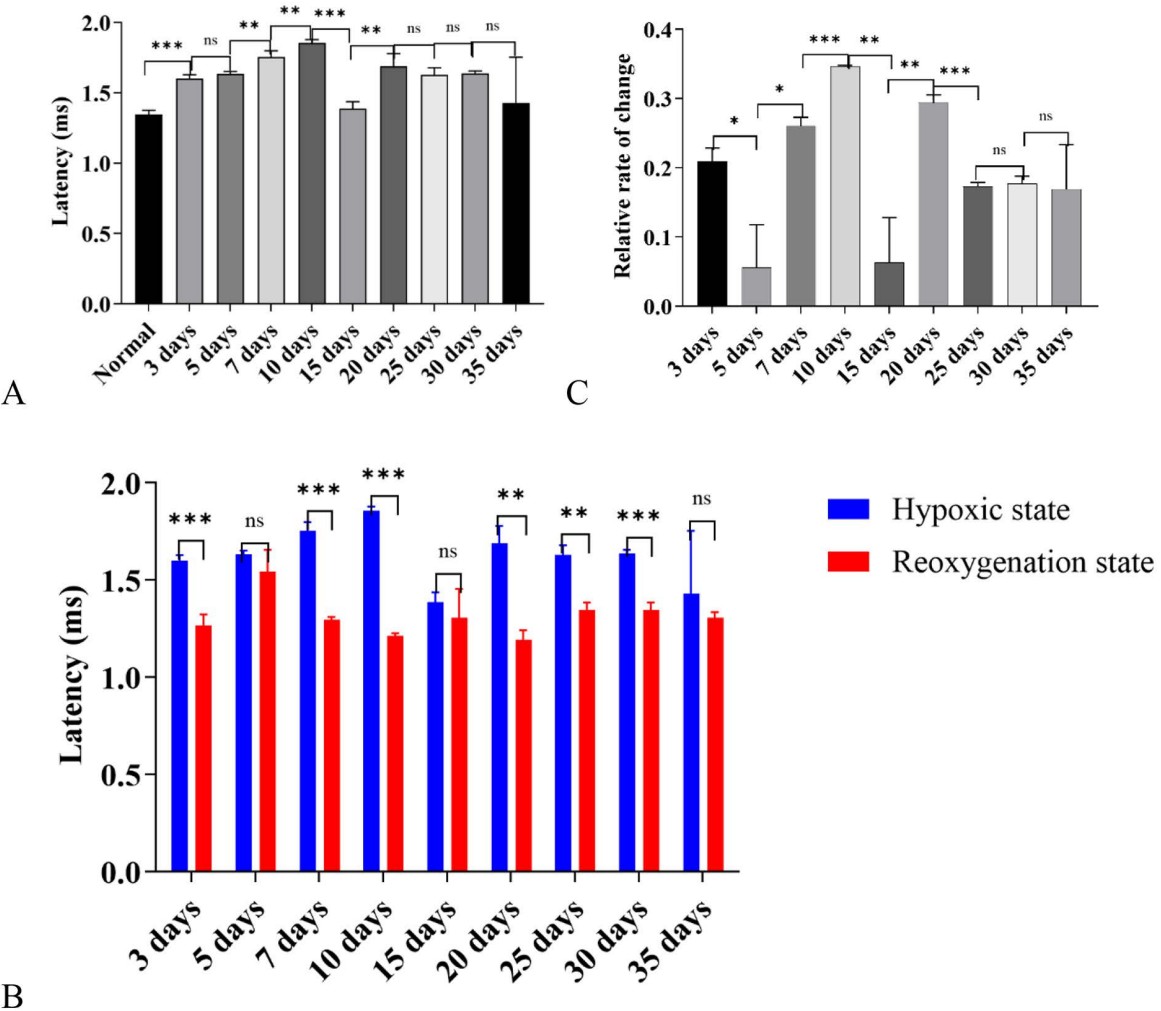

**Fig 14. Distribution of Compound Action Potential (AP) Latency Under Exposure to the High-Altitude Hypobaric and Hypoxic Environment. A:** This panel displays the distribution of AP latency across each group. Compared with the control group, the AP latencies of the groups with 15-day and 35-day exposure to the hypobaric hypoxic environment showed no significant difference ($P > 0.05$); all other groups exhibited a significant prolongation in AP latency, with the 7-day group and 10-day group showing the most obvious prolongation effect. **B:** This panel presents the changes in AP latency of each group before and after repressurization-reoxygenation **(R-R)**. **C:** This panel shows the relative change rate of each group before and after R-R (ns: $P > 0.05$, *: $P \leq 0.05$, **: $P < 0.01$, ***: $P < 0.001$).

**Results of -SP/AP area ratio.** Under the high-altitude hypobaric and hypoxic environment, the -SP/AP area ratios of the 5-day, 7-day, 25-day, and 30-day groups were higher than that of the control group, while those of the 10-day, 15-day, 20-day, and 35-day groups were lower than that of the control group. Among all groups, the 7-day group was the only one with an -SP/AP area ratio > 2.0, which was considered abnormal (Fig 15).

**Results of -SP/AP amplitude ratio.** Under the high-altitude hypobaric and hypoxic environment, analysis of the -SP/AP amplitude ratios across groups with different exposure durations revealed the following: The -SP/AP amplitude ratios of the 3-day, 5-day, 15-day, 20-day, 25-day, and 30-day groups were all > 0.4, suggesting the presence of endolymphatic hydrops in these groups; the -SP/AP amplitude ratios of the normal control group, 7-day group, 10-day group, and 35-day group were all < 0.4. Among these, the -SP/AP amplitude ratios of the 7-day group and 10-day group were close, and the values of both were approximately 50% of that of the normal control group (Fig 16).

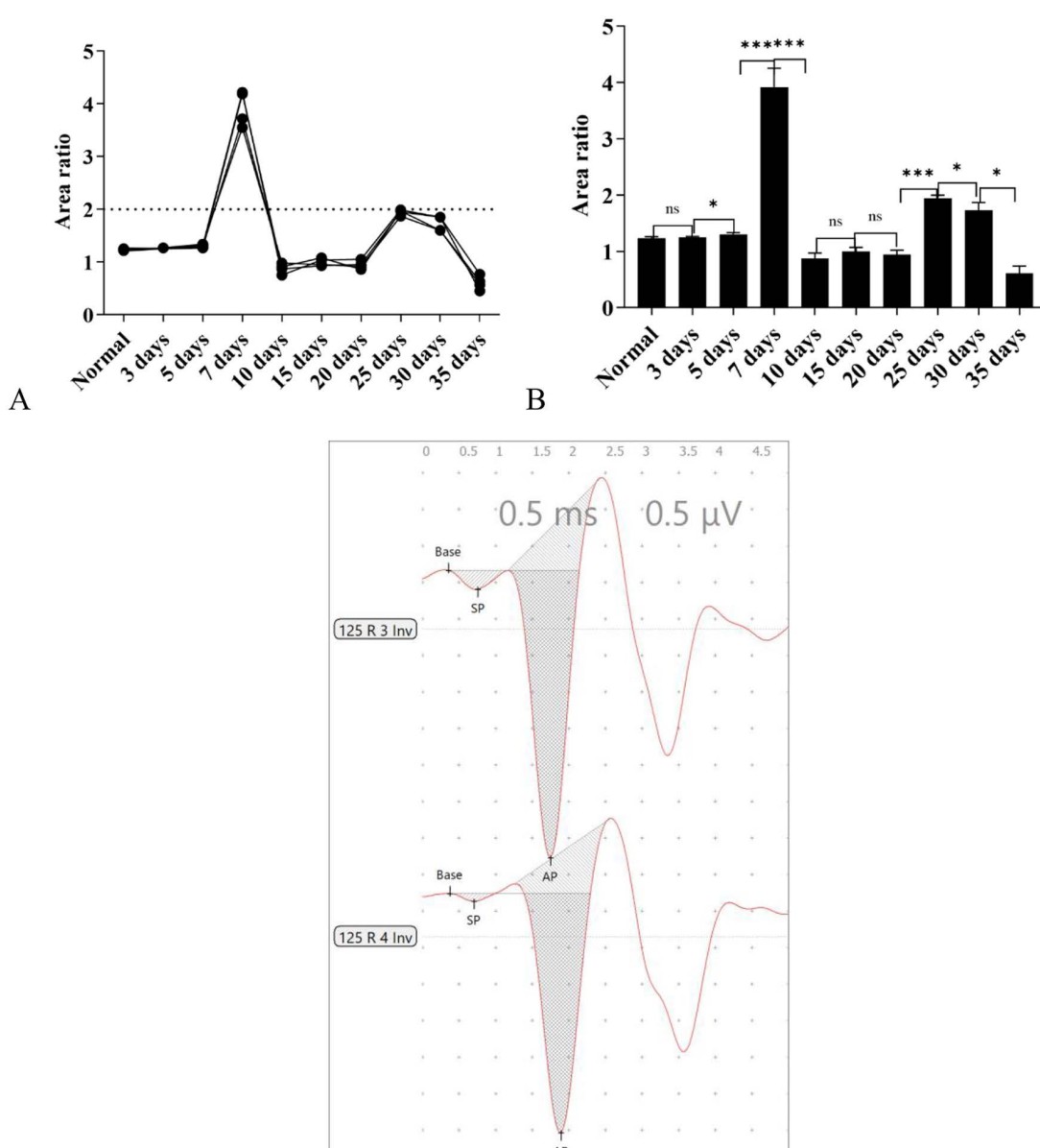

**Fig 15. A. This panel shows the overall trend of changes in the -SP/AP area ratio across each group.** B: This panel presents the specific comparison of -SP/AP area ratios across groups. It can be observed that the 7-day group had the highest -SP/AP area ratio, followed by the 25-day group and 30-day group, with the 7-day group having an -SP/AP area ratio > 2.0. In terms of group correlation, the groups with a significant increase in the -SP/AP area ratio were consistent with the previously observed group characteristics of "predominantly prolonged -SP latency". C: This panel illustrates the annotation method for the areas of -SP and AP in electrocochleography (EcochG) (ns: $P > 0.05$, *: $P \le 0.05$, **: $P < 0.01$, ***: $P < 0.001$).

**Trends and characteristics of EcochG waveform changes.** Under stimulation of the high-altitude hypobaric and hypoxic environment, the overall manifestations of EcochG were as follows: elevation of the -SP waveform, widening of the -SP-AP compound waveform, and a decrease or increase in amplitude (Fig 17).

**Summary of EcochG (electrocochleography) results.** The latency of the negative summating potential (-SP) demonstrated the highest sensitivity to stimulation under the high-altitude hypobaric hypoxic (HBH) environment.

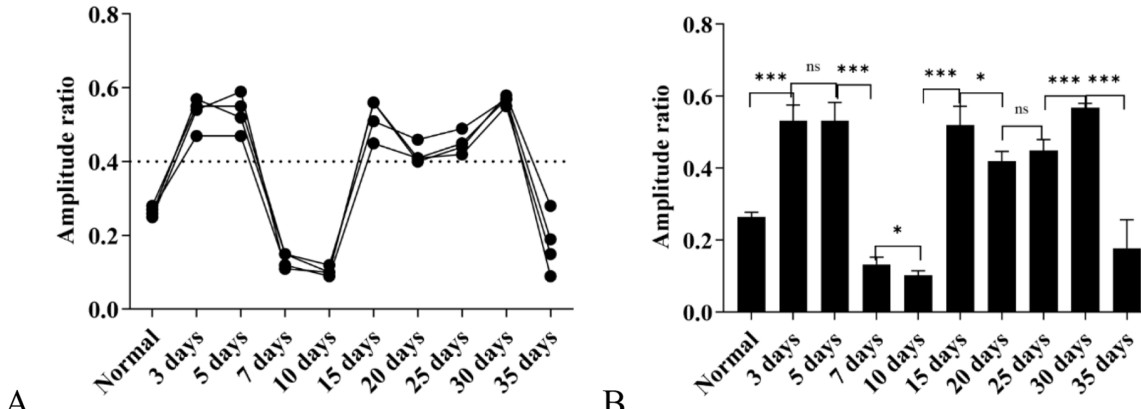

**Fig 16. Effects of different durations of high-altitude hypobaric hypoxia exposure on the -SP/AP amplitude ratio. A:** This panel shows the overall trend of changes in the -SP/AP amplitude ratio across each group under exposure to the high-altitude hypobaric and hypoxic environment. **B:** This panel clearly presents the distribution of -SP/AP amplitude ratios across each group. The -SP/AP amplitude ratio of the 35-day group showed no statistically significant difference from that of the normal control group (*P* > 0.05); the groups with an -SP/AP amplitude ratio > 0.4 were the 3-day, 5-day, 15-day, 20-day, 25-day, and 30-day groups; the groups with an -SP/AP amplitude ratio lower than that of the normal control group were the 7-day group and the 10-day group (*P* < 0.05) (ns: *P* > 0.05, *: *P* ≤ 0.05, **: *P* < 0.01, ***: *P* < 0.001).

Since -SP primarily reflects inner hair cell function, it can serve as a sensitive indicator of cochlear inner hair cell integrity.

Analysis of the -SP latency, compound action potential (AP) latency, and -SP/AP area ratio revealed that the 7th day represented the time point with the most pronounced and consistent response to HBH stimulation, whereas by the 35th day, all indicators had returned to control levels. This pattern closely paralleled the auditory brainstem response (ABR) latency trend. However, a key difference was observed: in EcochG, all three indicators declined to or below baseline by the 15th day, whereas the ABR threshold normalized earlier, on the 10th day. This temporal discrepancy suggests that, in clinical practice, consolidation therapy should continue even after apparent recovery of hearing function, to prevent potential relapse or delayed hearing loss.

Compared with -SP latency, the -SP/AP area ratio showed lower sensitivity but higher specificity to HBH-induced auditory stress.

Interestingly, the -SP/AP amplitude ratio exhibited a distinct pattern: in the 7-day group, the ratio was < 0.4, approximately 50% of that in controls. According to the conventional diagnostic criterion that an -SP/AP amplitude ratio > 0.4 indicates endolymphatic hydrops, the 7-day group would be classified as normal. However, this finding contradicts the trends observed in most ABR and EcochG parameters, even showing an opposite direction of change. This raises a critical question: should an abnormally low -SP/AP amplitude ratio also be considered pathological? This hypothesis warrants further experimental validation.

## DPOAE (distortion product otoacoustic emissions) results

A total of 40 C57BL/6J mice were included in the DPOAE testing in this study. Among them, only 3 mice had "passed" DPOAE test results, while the remaining 37 mice had "failed" results. The overall pass rate of the DPOAE test for this batch of mice was 7.5%. It should be specifically noted that the 3 mice with "passed" DPOAE test results were all from the normal control group.

Based on the above test results, regarding the core scientific question of "whether the high-altitude hypobaric and hypoxic environment affects the function of cochlear outer hair cells", the current volume of experimental data and sample size in this study are insufficient to draw a clear conclusion, and a definitive answer cannot be provided temporarily Fig 18.

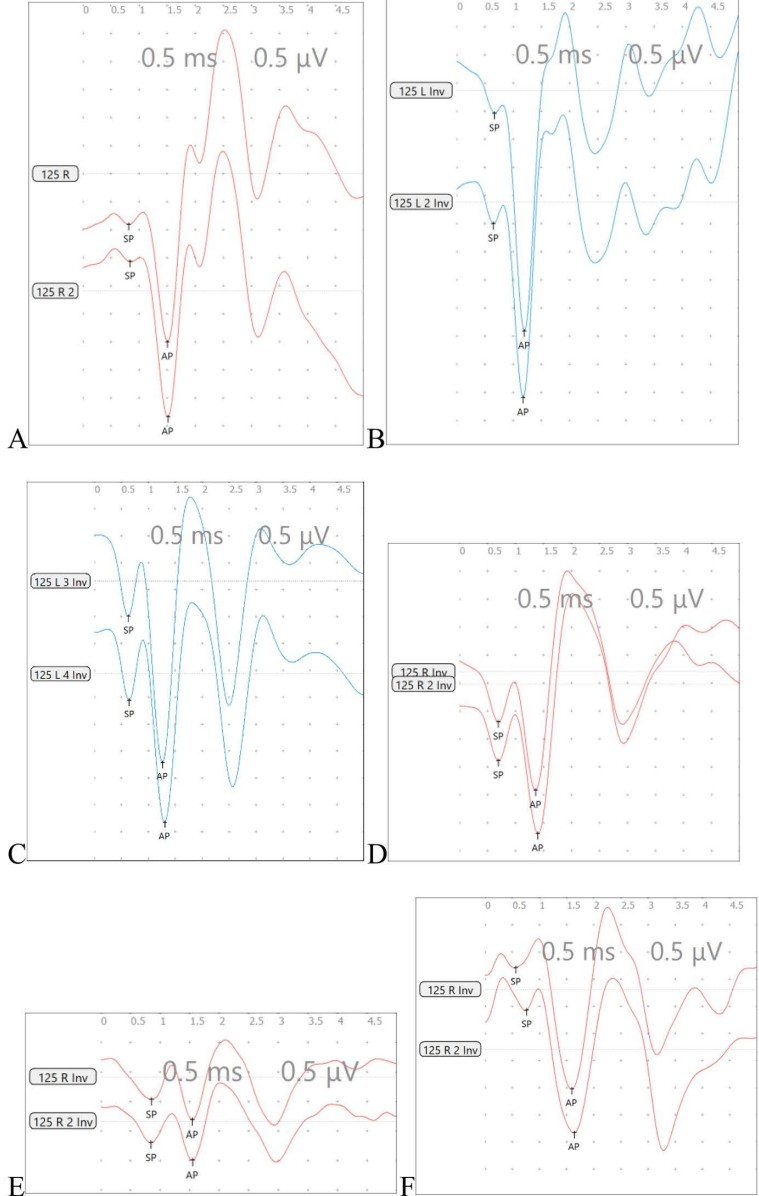

**Fig 17. The Change Trends of EcochG Under Exposure to the High-Altitude Hypobaric and Hypoxic Environment. A:** This panel shows the EcochG of the right ear from a C57BL/6J mouse in the normal control group. **B and C:** These panels show an increase and elevation of -SP in the left ear. **D:** This panel shows elevation of -SP and widening of the waveform in the EcochG of the right ear. **E:** This panel shows a decrease in the amplitudes of -SP and AP, as well as widening of the waveform, in the EcochG of the right ear. **F:** This panel shows an increase in the amplitudes of -SP and AP at the late stage of exposure to the high-altitude hypobaric and hypoxic environment, while the waveform remains relatively wide.

## Discussion

As a key pathological feature of Ménière's disease, endolymphatic hydrops may play a significant role in the development of hearing loss induced by high-altitude hypoxia. The pathological mechanisms it mediates and its effects on physiological functions offer a valuable complementary perspective for interpreting the results of this study. Previous research has demonstrated that the mechanical pressure caused by endolymphatic hydrops can directly act on the membranous

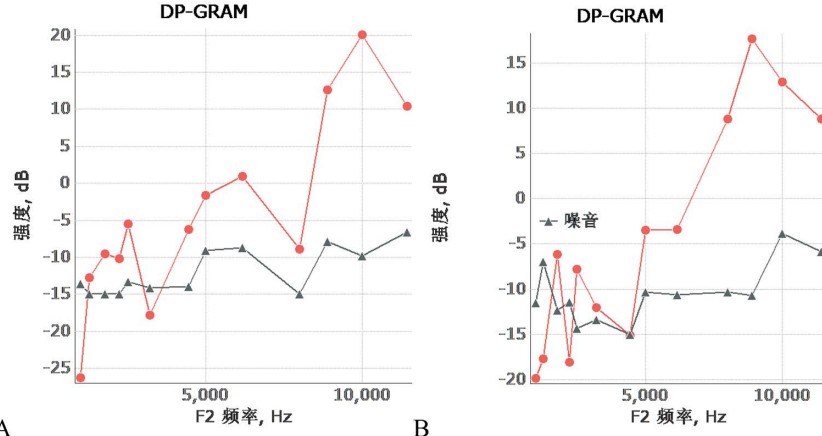

**Fig 18. DPOAE (Distortion Product Otoacoustic Emissions) Test Results of C57BL/6J Mice Under Exposure to the High-Altitude Hypobaric and Hypoxic Environment. A:** This panel shows the DPOAE test result: "Passed". **B:** This panel shows the DPOAE test result: "Failed".

labyrinth of the cochlea, altering the elastic properties and mass loading of the basilar membrane. These changes lead to sensorineural hearing loss and diplacusis in the apical region of the cochlea, which encodes low-frequency sounds. This phenomenon is closely associated with the shift in the site of maximum displacement of the traveling (von Békésy) wave along the basilar membrane and differences in its intrinsic stiffness [17–19].

Importantly, hypoxic environments can markedly aggravate cochlear damage in hydropic ears and produce distinct effects on auditory nerve responses to high- and low-intensity pure tones, suggesting a potential synergistic detrimental interaction between hypoxia and endolymphatic hydrops. This finding provides critical mechanistic insight into hearing loss under high-altitude hypoxic conditions [20].

From the perspective of physiological correlates, the extent of endolymphatic hydrops shows a significant positive correlation with the summating potential (SP) amplitude detected by electrocochleography. It can also induce saccular dysfunction, as evidenced by increased cVEMP thresholds, reduced amplitudes, or waveform absence, and suppress the modulatory function of low-frequency distortion product otoacoustic emissions (DPOAEs). These alterations not only confirm the direct impact of hydrops on the auditory conduction pathway but also suggest potential biomarkers for assessing the pathological progression of hypoxia-related hearing impairment [18].

Furthermore, the strong association between endolymphatic hydrops and fluctuating sensorineural hearing loss, as well as the reversible recovery of auditory indicators following dehydration therapy, implies that interventions targeting hydrops may offer a novel therapeutic avenue for preventing and treating high-altitude hypoxia-induced hearing damage [18,20].

Overall, the integration of these mechanistic and clinical observations underscores the potential mediating role of endolymphatic hydrops in hypoxia-related auditory pathology. It also bridges the gap between animal experimental findings and human physiological mechanisms, thereby enhancing the translational significance and clinical applicability of this study.

## Effects of high-altitude hypobaric hypoxia on ABR: Nerve conduction velocity, threshold, and frequency specificity

This study found that after exposure of C57BL/6J mice to a high-altitude hypobaric hypoxic (HBH) environment, the latencies of all ABR waves under click, 4000 Hz, and 8000 Hz tone-burst stimulation were generally prolonged, with the most pronounced delay observed in Wave II (originating from the cochlear nucleus). This finding indicates that HBH exposure slows postsynaptic neural conduction along the cochlea-inferior colliculus pathway, and suggests that the cochlear nucleus may represent a key hypoxia-sensitive site.

In addition to latency changes, the ABR thresholds were significantly elevated after HBH exposure, confirming functional impairment of the auditory pathway. The combination of threshold elevation and latency prolongation demonstrates that hypobaric hypoxia induces hearing dysfunction through neural conduction damage.

From a temporal perspective, ABR indicators exhibited distinct time dependence. The 7-day group showed the most stable and pronounced latency prolongation, while its threshold increase, though significant, was less marked. This discrepancy implies the presence of a negative feedback regulatory mechanism within the auditory system that mitigates the cumulative effects of hypoxia-induced latency delay, thereby limiting functional deterioration and providing a degree of neuroprotective regulation.

The central auditory feedback network exerts precise modulation over downstream auditory processing, extending its influence from the brainstem to peripheral structures such as the cochlea. Notably, compensatory activity at the cochlear nucleus level can influence waveform morphology, amplitude, and latency of auditory evoked potentials (including ABR and auditory steady-state responses) via neural feedback pathways. This underscores the multilevel and integrated nature of auditory system regulation [21].

Regarding frequency specificity, the click stimulus-a transient 100 μs broadband signal-lacks frequency selectivity but synchronously activates hair cells and auditory nerve fibers, primarily reflecting the 2–4 kHz range [8,22]. Given that the mouse auditory range spans 0.2–10 kHz, both 4000 Hz and 8000 Hz stimuli effectively assess low- to mid-frequency hearing.

This study demonstrated that HBH exposure predominantly induces low-frequency hearing loss in C57BL/6J mice, consistent with EcochG findings. Mechanistically, cochlear hair cells, which convert acoustic energy into electrical signals, connect directly to the central auditory system via spiral ganglion neurons. The axons of these neurons form the auditory nerve, which projects to the cochlear nucleus complex, comprising the anterior ventral (AVCN), posterior ventral (PVCN), and dorsal (DCN) cochlear nuclei. The ventral cochlear nucleus, particularly innervated by fibers from the basal turn of the cochlea, is responsible for low-frequency sound perception [23–26]. This anatomical feature provides mechanistic support for the observed low-frequency hearing loss induced by HBH exposure.

### Time dependence of ABR waves and their conduction regulatory mechanisms

Further analysis of the time-dependent changes in ABR waves reveals the regulatory pattern of auditory conduction under high-altitude hypobaric hypoxia (HBH). Specifically, Wave I (cochlear nerve) latency showed continuous prolongation from days 3–20, peaking on day 15, suggesting slowed cochlea-cochlear nerve conduction. From day 25 onward, latency returned to near-normal levels, indicating a self-repair capacity of the cochlear nerve against hypoxic damage.

Wave II (cochlear nucleus) latency began to increase on day 5 and peaked on day 7, implying a marked slowdown in neural transmission at this site. By day 35, latency became shorter than in controls, suggesting an adaptive response to hypoxia. Amplitude changes revealed stage-specific mechanisms: the 5-day group showed increased amplitude (higher voltage) with prolonged latency-indicating increased resistance; the 7-day group exhibited unchanged amplitude but prolonged latency, also consistent with elevated resistance. In contrast, amplitudes decreased in the 10-, 20-, and 25-day groups, implying that latency prolongation during this phase was primarily driven by reduced voltage rather than resistance changes.

Wave III (superior olivary complex) showed a similar trend, with the most pronounced latency prolongation on days 7 and 10, followed by a decline below control levels on day 35, reflecting adaptive neural adjustment. The 3-, 7-, 20-, and 30-day groups exhibited amplitude increases (with prolonged latency), again suggesting elevated resistance, whereas the 35-day group showed decreased amplitude and shortened latency-likely reflecting optimized conduction efficiency at the superior olivary complex. The oscillatory amplitude pattern further indicates that auditory adaptation to hypoxia is a gradual, fluctuating process.

Wave IV (lateral lemniscus) was less affected, showing notable latency prolongation only on day 30, suggesting lower hypoxia sensitivity. In contrast, Wave V (inferior colliculus) displayed the greatest prolongation on days 7 and 10, followed by normalization from day 15, indicating that the inferior colliculus exhibits strong adaptive resilience to hypoxia.

Collectively, these findings suggest that HBH exposure induces time-dependent alterations in resistance and voltage across different segments of the auditory pathway, thereby modulating ABR latency and threshold and ultimately contributing to low-frequency hearing loss.

From a physiological perspective, temporal coding of acoustic signals begins at the auditory brainstem [21]. The temporal precision of brainstem responses is closely linked to speech perception [27,28] and reading ability [29], reflecting early-stage neuronal encoding of temporal structure [30, 26]. Moreover, sound temporal regularity modulates neuronal firing rates in the cochlear nucleus (CN) [31]. Regarding amplitude regulation, the medial olivocochlear reflex (MOCR) serves as a crucial efferent feedback mechanism that modulates peripheral auditory activity. Studies have shown that medial olivocochlear (MOC) neurons can either enhance or suppress neuronal firing depending on the intensity of the incoming acoustic-electrical signal [21,32].

### EcochG findings: Diagnostic controversies and mechanisms of endolymphatic hydrops

EcochG primarily records the compound wave of the summating potential (SP) and action potential (AP) via click stimulation. Among its parameters, an -SP/AP amplitude ratio > 0.4 is a commonly used clinical indicator for diagnosing endolymphatic hydrops and loudness recruitment, while an -SP/AP area ratio > 2.0 is considered abnormal [9].

In this study, the -SP/AP amplitude ratio exceeded 0.4 in the 3-day, 5-day, 15-day, 20-day, 25-day, and 30-day groups exposed to the high-altitude hypobaric hypoxic environment, suggesting the presence of endolymphatic hydrops. However, the -SP/AP amplitude ratio in the 7-day group was only 50% of that in the control group (showing no indicator of endolymphatic hydrops). In contrast, the 7-day group was the only group with an -SP/AP area ratio exceeding 2.0, which should be regarded as abnormal according to standards-this result is consistent with the ABR findings of each group but contradicts the -SP/AP amplitude ratio results. Combined with the clinical research finding that "the sensitivity of an -SP/AP amplitude ratio > 0.4 for diagnosing Ménière's disease is only 60%" [9,33,12], this suggests that using "an -SP/AP amplitude ratio > 0.4" alone as the diagnostic criterion for endolymphatic hydrops may have limitations. Whether "a significantly lower-than-normal -SP/AP amplitude ratio" should be incorporated into abnormal indicators requires verification in studies with larger sample sizes.

Mechanistically, the stability of SP depends on the ionic homeostasis (high potassium and low sodium levels) of cochlear endolymph and the endolymphatic potential (+80 mV), and this homeostasis is maintained by the stria vascularis. During endolymphatic hydrops, the ionic balance is disrupted, which can lead to dysfunction of inner hair cells and induce SP abnormalities: On one hand, increased endolymph pressure impairs the mechano-electrical transduction process of inner hair cells, resulting in distortion of sound responses and manifested as a relative increase in SP amplitude and width; on the other hand, the high-pressure environment may inhibit the activity of auditory nerve afferent fibers, leading to a decrease in AP amplitude. Ultimately, these changes cause an elevation in the -SP/AP ratio [5,6,9]. This mechanism also explains the hypobaric hypoxia-induced low-frequency hearing loss observed in this study-endolymphatic hydrops is a common pathological basis for low-frequency hearing loss, consistent with the pathological characteristics of Ménière's disease and low-frequency sudden sensorineural hearing loss [34–38].

### Limitations and reflections on the research results

Distortion Product Otoacoustic Emissions (DPOAE) serve as a specific indicator for evaluating the active motile function of cochlear outer hair cells. Abnormalities in this measure typically suggest outer hair cell dysfunction, manifested as reduced auditory sensitivity. In this study, no valid DPOAE signals were obtained from any mice in the experimental groups, a result that warrants cautious interpretation.

On one hand, this phenomenon may be partially attributable to statistical bias arising from a limited sample size. On the other hand, previous studies have demonstrated that activation of the medial olivocochlear reflex (MOCR) in animal models can significantly suppress stimulus-evoked peripheral auditory responses, including otoacoustic emissions [39–44]. This suggests that the absence of detectable DPOAE signals in the present study may result either from direct damage to cochlear outer hair cells induced by high-altitude hypobaric hypoxia, or from MOCR activation under such environmental conditions, which in turn inhibits DPOAE responses. These two possibilities require further investigation.

Moreover, this study only performed a comparative analysis between mid-altitude and high-altitude regions, without including a simultaneous lowland (sea-level) control group. Although reference data from plain areas were cited for comparison, it remains difficult to fully exclude the potential influence of regional environmental factors on data interpretation.

To elucidate the underlying mechanisms by which hypobaric hypoxia affects cochlear outer hair cell function, future studies should aim to increase the sample size to minimize statistical bias and incorporate morphological assessments of outer hair cells. Establishing a dual verification framework that integrates both functional and structural analyses will ultimately help clarify the regulatory mechanisms of high-altitude hypobaric hypoxia on cochlear outer hair cell physiology.

### Outstanding questions to be addressed

In China, hyperbaric oxygen therapy (HBOT) is a commonly used clinical method to improve inner ear ischemia and hypoxia. However, for patients who have developed low-frequency hearing loss due to exposure to high-altitude hypobaric hypoxia, after receiving HBOT, they need to be re-exposed to the high-altitude hypobaric hypoxic environment upon exiting the hyperbaric oxygen chamber. A key question arises: Will this process - "exposure to hypobaric hypoxic environment - hyperbaric oxygen intervention - re-exposure to hypobaric hypoxic environment" – trigger a specific mechanism similar to "ischemia-reperfusion injury"? Whether this mechanism will exert a secondary impact on the already impaired low-frequency auditory function, thereby interfering with the recovery process of low-frequency hearing or causing long-term damage, remains unclear at present. To address this uncertainty, further targeted animal experiments and clinical follow-up studies are required to obtain a definitive answer.

### Conclusions

Exposure to a high-altitude hypobaric hypoxic environment can induce low-frequency hearing loss in C57BL/6J mice, and the mice's responsive regulation or adaptation to hypobaric hypoxic stimulation exhibits time dependence.

The low-frequency hearing loss induced by exposure to a high-altitude hypobaric hypoxic environment is closely associated with the slowed transmission speed of postsynaptic electrical signals in the cochlea-inferior colliculus auditory conduction pathway.

The slowed transmission speed of postsynaptic electrical signals in the auditory conduction pathway, caused by exposure to a high-altitude hypobaric hypoxic environment, may be related to changes in voltage or resistance at various sites (e.g., the cochlear nucleus) during acoustic signal conduction.

### Supporting information

**S1 File. English version of mouse audiometric data.**
(ZIP)

### Acknowledgments

The smooth launch and successful completion of this study were made possible through the generous support and enthusiastic assistance of many teachers and colleagues. We express our sincere gratitude and highest respect to all of them.

We are especially grateful to Director Zhang Ying and Associate Director Guo Bin of the Department of Otolaryngology, Affiliated Hospital of Qinghai University, for providing essential experimental equipment, including surgical microscopes, and for offering valuable guidance and suggestions in refining research protocols, advancing experiments, and addressing key challenges-contributions that laid a solid foundation for the smooth progress of this study.

We also extend heartfelt thanks to Senior Sister Zhang Qingping of Qinghai University for her meticulous guidance and dedicated assistance during the experimental procedures. Her rigorous academic attitude and selfless support greatly ensured the study's efficiency and success.

Finally, we sincerely thank all institutions and individuals who have offered their support and assistance throughout this research.

## Author contributions

**Conceptualization:** yi wang.

**Data curation:** Benhong Ren, Qingping Zhang, Wenyuan Gan.

**Formal analysis:** yi wang, Qingping Zhang, Ying Zhang.

**Funding acquisition:** yi wang.

**Investigation:** Benhong Ren, Qingping Zhang.

**Methodology:** yi wang, Shanhong Li, Xiaoli Zhang.

**Project administration:** yi wang.

**Resources:** yi wang, Wenyuan Gan, Guanghao Yue, Shanhong Li, Xiaoli Zhang, Wenjun Cao, Feng Tang, Ying Zhang, Bin Guo.

**Software:** Benhong Ren, Qingping Zhang.

**Supervision:** yi wang, Ying Zhang.

**Validation:** Benhong Ren, Qingping Zhang.

**Visualization:** Benhong Ren.

**Writing – original draft:** Benhong Ren.

**Writing – review & editing:** yi wang.

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
