## [Decision Letter · Decision Letter 0]

9 Dec 2025

Dear Dr. wang,

Thank you for submitting your manuscript to PLOS ONE. After careful consideration, we feel that it has merit but does not fully meet PLOS ONE’s publication criteria as it currently stands. Therefore, we invite you to submit a revised version of the manuscript that addresses the points raised during the review process.

**ACADEMIC EDITOR:**

We look forward to receiving your revised manuscript.

Kind regards,

Gauri Mankekar, MD,PhD,FACS

Academic Editor

PLOS One

2. Please note that your Data Availability Statement is currently missing [the repository name and/or the DOI/accession number of each dataset OR a direct link to access each database]. If your manuscript is accepted for publication, you will be asked to provide these details on a very short timeline. We therefore suggest that you provide this information now, though we will not hold up the peer review process if you are unable.

Additional Editor Comments (if provided):

Reviewers' comments:

Reviewer's Responses to Questions

**Comments to the Author**

1. Is the manuscript technically sound, and do the data support the conclusions?

Reviewer #1: Partly

Reviewer #2: Yes

2. Has the statistical analysis been performed appropriately and rigorously?

Reviewer #1: Yes

Reviewer #2: Yes

3. Have the authors made all data underlying the findings in their manuscript fully available?

Reviewer #1: Yes

Reviewer #2: Yes

4. Is the manuscript presented in an intelligible fashion and written in standard English?

Reviewer #1: Yes

Reviewer #2: Yes

Reviewer #1: Thank you for the opportunity to review this interesting article. The methods and results appear very thorough. One area where there may be some enhancement is the potential link with endolymphatic hydrops. It might be helpful to expand on this a little to help the reader understand and how this potentially compares or translates with human physiology.

Reviewer #2: The manuscript explores how simulated high-altitude hypobaric hypoxia affects low-frequency hearing in mice using ABR, ECochG, and DPOAE. I have the following concerns;

-The study uses only 4 mice per subgroup. The small n per group raises concerns about statistical power and the generalizability of findings.

-The conclusion suggests that: “the slowing of postsynaptic electrical signal transmission may be related to changes in voltage or resistance… likely associated with endolymphatic hydrops.”This claim lacks direct evidence; ECochG SP/AP ratios alone are not sufficient to diagnose hydrops in mice.

-The control group was housed at a moderate altitude (2260 m), not sea level. This means the "control" itself is exposed to some level of hypobaric hypoxia, which may attenuate observed effects and complicate interpretation.

-The manuscript is over 10,000 words, contains repeated explanations (e.g., ABR wave origins, ECochG physiology), and includes long figure captions that restate results.

-The discussion is long, descriptive, and does not sufficiently address study limitations.

**Do you want your identity to be public for this peer review?** For information about this choice, including consent withdrawal, please see our Privacy Policy

Reviewer #1: No

Reviewer #2: No

---

## [Author Response · Author response to Decision Letter 1]

23 Dec 2025

Response to Reviewers

1. Is the academic methodology rigorous? Do the data support the conclusions?

The paper must adopt rigorous scientific research methods, and the data must fully support the conclusions; the experimental design should be standardized, including reasonable controls, repeated experiments, and an adequate sample size, with conclusions logically derived based on the presented data.

• Reviewer 1: Partially compliant

• Reviewer 2: Compliant

2. Is the statistical analysis appropriate and rigorous?

• Reviewer 1: Compliant

• Reviewer 2: Compliant

3. Have the authors fully disclosed all raw data related to the research?

In accordance with PLOS ONE’s data policy, except for very few exceptions, authors must unrestrictedly disclose all raw data relevant to the research conclusions (see the Data Availability Statement in the paper’s PDF). Data can be submitted as part of the main manuscript, supplementary materials, or uploaded to a public database. For example, in addition to providing statistical summaries, raw data points corresponding to statistics such as mean, median, and variance must be disclosed. If there are restrictions on data sharing (e.g., involving research participant privacy, use of third-party data, etc.), the nature of the restrictions must be clearly stated.

• Reviewer 1: Compliant

• Reviewer 2: Compliant

4. Is the paper clearly expressed? Does the language conform to standard English norms?

PLOS ONE does not provide manuscript editing or proofreading services. Submitted papers must be clear, accurate, and unambiguous. All spelling and grammatical errors should be corrected during the revision stage; if there are specific errors, please indicate them here.

• Reviewer 1: Compliant

• Reviewer 2: Compliant

5. Detailed Reviewer Comments and Author Responses

Reviewer 1’s Comments

Thank you for the opportunity to review this valuable paper. The research methods and results are presented in great detail. An area for further improvement is to supplement the discussion on the potential association with endolymphatic hydrops; appropriate elaboration will help readers understand its similarities, differences, and translational value relative to human physiological mechanisms.

Response: We appreciate the reviewer's recognition of the study's value and constructive suggestions. We have systematically reviewed literature related to endolymphatic hydrops and, combined with the core results of this study, specifically added discussions on its potential associations in the first paragraph of the Discussion section. We focus on sorting out the similarities, differences, and potential translational value relative to human physiological mechanisms to help readers fully understand the research significance.

Reviewer 2’s Comments

This study investigates the effects of simulated high-altitude hypobaric hypoxia on low-frequency hearing in mice using Auditory Brainstem Response (ABR), Electrocochleography (ECochG), and Distortion Product Otoacoustic Emissions (DPOAE) techniques. I have the following questions:

1.Only 4 mice were used per subgroup, resulting in a small sample size that may affect statistical power and the generalizability of the research results.

Response: We appreciate the reviewer's valuable comment. The sample size of 4 mice per subgroup was mainly constrained by the operational limitations of hypobaric hypoxia modeling: the duration of experimenters’ entry into the modeling chamber must be strictly controlled, as prolonged stay would disrupt the stability of the chamber environment and thus affect modeling effects; meanwhile, the time window for reoxygenation must be precisely regulated, making it difficult to include more experimental animals in a single experiment. We fully agree that a small sample size may have a certain impact on statistical power and the generalizability of the results. In future studies, we will expand the sample size by increasing the number of experimental batches and expanding the research team to further improve the reliability and generalizability of the research results. Thank you again for the insightful suggestion.

2.The conclusion states that "the slowing of postsynaptic electrical signal conduction may be related to changes in voltage or resistance... and may be associated with endolymphatic hydrops," but this claim lacks direct evidence-ECochG SP/AP ratio alone is insufficient to diagnose endolymphatic hydrops in mice.

Response: We appreciate the reviewer's pertinent comment. We fully acknowledge that this inference lacks direct evidence and that the ECochG SP/AP ratio alone is insufficient to diagnose endolymphatic hydrops in mice. Therefore, we have deleted the relevant inference stating that "the slowing of postsynaptic electrical signal conduction may be related to changes in voltage or resistance... and may be associated with endolymphatic hydrops" in the conclusion section to ensure that the conclusions are rigorous and within the scope supported by experimental evidence.

3.The control group was housed at a moderate altitude (2260 meters) rather than sea level, meaning the control group itself was exposed to a certain degree of hypobaric hypoxia, which may weaken the observed effect intensity and increase the complexity of result interpretation.

Response: We appreciate the reviewer's valuable comment. We agree that housing the control group at a moderate altitude (2260 meters) instead of sea level may weaken the observed effect intensity and increase the complexity of result interpretation. We have supplemented and cited research data from plain areas (sea level) for comparative analysis in the paper, and added a dedicated "Limitations" section in the Discussion to elaborate on the potential influencing factors related to this experimental design, ensuring the comprehensiveness and rigor of result interpretation.

4.The paper exceeds 10,000 words, with redundant content (e.g., the origin of ABR waveforms, ECochG physiological mechanisms) and lengthy figure legends that repeat result descriptions.

Response: We appreciate the reviewer's precise comment. We have systematically optimized the paper for excessive length and redundant content: first, deleted redundant elaborations such as the origin of ABR waveforms and ECochG physiological mechanisms in the first paragraph of the Discussion; second, streamlined the figure legends by removing repeated result descriptions to ensure they are concise, accurate, and focused, effectively reducing the paper length while enhancing content logic.

5.The Discussion section is overly long, descriptive in nature, and fails to fully address the study’s limitations.

Response: We appreciate the reviewer's reminder. We have added a limitation statement regarding "the absence of a plain-area control group" in the Discussion as required, and deleted redundant descriptions to strengthen the pertinence and depth of the Discussion.

---

## [Decision Letter · Decision Letter 1]

21 Jan 2026

Exposure to high-altitude hypobaric hypoxic environment induces low-frequency hearing loss in C57BL/6J mice: mediated by slowing down the postsynaptic electrical signal transmission speed in the cochlear-inferior colliculus auditory signaling pathway

PONE-D-25-56186R1

Dear Dr. Wang,

We’re pleased to inform you that your manuscript has been judged scientifically suitable for publication and will be formally accepted for publication once it meets all outstanding technical requirements.

Kind regards,

Gauri Mankekar, MD,PhD,FACS

Academic Editor

PLOS One

Additional Editor Comments (optional):

Reviewers' comments:

Reviewer's Responses to Questions

**Comments to the Author**

Reviewer #2: All comments have been addressed

2. Is the manuscript technically sound, and do the data support the conclusions?

Reviewer #2: Yes

3. Has the statistical analysis been performed appropriately and rigorously?

Reviewer #2: Yes

4. Have the authors made all data underlying the findings in their manuscript fully available?

Reviewer #2: Yes

5. Is the manuscript presented in an intelligible fashion and written in standard English?

Reviewer #2: Yes

Reviewer #2: I am satisfied with authors responses and corrections in the revised version of the manuscript. No further comments.

**Do you want your identity to be public for this peer review?** For information about this choice, including consent withdrawal, please see our Privacy Policy

Reviewer #2: **Yes:** Firas Alzoubi

---

## [Editor Report · Acceptance letter]

PONE-D-25-56186R1

PLOS One

Dear Dr. wang,

I'm pleased to inform you that your manuscript has been deemed suitable for publication in PLOS One. Congratulations! Your manuscript is now being handed over to our production team.

Kind regards,

on behalf of

Dr. Gauri Mankekar

Academic Editor

PLOS One